# Wave-particle energy exchange directly observed in a kinetic Alfvén-branch wave

Daniel J. Gershman[1,2], Adolfo F-Viñas[2], John C. Dorelli[2], Scott A. Boardsen[2,3], Levon A. Avanov[1,2], Paul M. Bellan[4], Steven J. Schwartz[5], Benoit Lavraud[6,7], Victoria N. Coffey[8], Michael O. Chandler[8], Yoshifumi Saito[9], William R. Paterson[2], Stephen A. Fuselier[10], Robert E. Ergun[11], Robert J. Strangeway[12], Christopher T. Russell[12], Barbara L. Giles[2], Craig J. Pollock[2], Roy B. Torbert[13,14] & James L. Burch[10]

Alfvén waves are fundamental plasma wave modes that permeate the universe. At small kinetic scales, they provide a critical mechanism for the transfer of energy between electromagnetic fields and charged particles. These waves are important not only in planetary magnetospheres, heliospheres and astrophysical systems but also in laboratory plasma experiments and fusion reactors. Through measurement of charged particles and electromagnetic fields with NASA's Magnetospheric Multiscale (MMS) mission, we utilize Earth's magnetosphere as a plasma physics laboratory. Here we confirm the conservative energy exchange between the electromagnetic field fluctuations and the charged particles that comprise an undamped kinetic Alfvén wave. Electrons confined between adjacent wave peaks may have contributed to saturation of damping effects via nonlinear particle trapping. The investigation of these detailed wave dynamics has been unexplored territory in experimental plasma physics and is only recently enabled by high-resolution MMS observations.

[1] Department of Astronomy, University of Maryland, College Park, Maryland 20742, USA. [2] NASA Goddard Space Flight Center, Greenbelt, Maryland 20771, USA. [3] Goddard Planetary Heliophysics Institute, University of Maryland, Baltimore County, Maryland 21250, USA. [4] Division of Engineering and Applied Science, California Institute of Technology, Pasadena, California 91125, USA. [5] Blackett Laboratory, Imperial College London, London SW7 2AZ, UK. [6] Institut de Recherche en Astrophysique et Planétologie, Université de Toulouse, Toulouse F-31400, France. [7] Centre National de la Recherche Scientifique, UMR 5277, Toulouse F-31400, France. [8] NASA Marshall Space Flight Center, Huntsville, Alabama 35808, USA. [9] JAXA Institute of Space and Astronautical Science, Sagamihara, Kanagawa 252-5210, Japan. [10] Southwest Research Institute, San Antonio, Texas 78238, USA. [11] Astrophysical and Planetary Sciences, University of Colorado, Boulder, Colorado 80305, USA. [12] Department of Earth, Planetary, and Space Sciences, University of California, Los Angeles, California 90095, USA. [13] Physics Department, University of New Hampshire, Durham, New Hampshire 03824, USA. [14] Southwest Research Institute Durham, Durham, New Hampshire 03824, USA. Correspondence and requests for materials should be addressed to D.J.G. (email: daniel.j.gershman@nasa.gov).

The Alfvén wave is a ubiquitous plasma wave mode wherein ions collectively respond to perturbations in the ambient magnetic field direction[1]. No net energy is transferred between the field and the plasma particles in ideal Alfvén waves. However, ion motion decouples from electron motion when wave dynamics are faster than ion orbital motion around the local magnetic field or are on scales smaller than the ion orbit size, defined by the gyrofrequency ($\omega_{ci}$) and gyroradius ($\rho_i$), respectively. When the perpendicular spatial scale of an Alfvén wave approaches $\rho_i$, the wave can support significant parallel electric and magnetic field fluctuations that enable net transfer of energy between the wave field and plasma particles via Landau or transit–time interactions[2–4].

The transition of an ideal fluid-scale Alfvén wave to a kinetic-scale Alfvén wave (KAW) occurs at $k_\perp \rho_i \sim 1$ and $k_\perp > k_\parallel$, where $\mathbf{k}$ is the wavevector and '$\perp$' and '$\parallel$' are defined with respect to the local magnetic field direction. These KAWs are essential for energy transfer processes in plasmas. Broadband KAWs have long been associated in space physics with turbulent heating in the solar wind and magnetosheath[5–7] and are also thought to account for a substantial amount of the energy input into Earth's auroral regions that can drive charged particle outflow and atmospheric loss[8–13]. In the laboratory, KAWs can transport energy away from the core regions of fusion plasmas, resulting in the unwanted deposition of energy at the reactor edges[14,15]. Understanding kinetic-scale wave generation, propagation and interaction with charged particles is critical to unraveling and predicting the relevant physics of these fundamental processes.

Alfvén wave theory predicts that transverse fluctuations in the current density ($\mathbf{J}$) and electron-pressure-gradient-driven electric field ($\mathbf{E}_p = -\nabla \bullet \underline{\mathbf{P}}_e / (n_e e)$) are 90° out of phase with one another, such that the plasma heating term, $\Delta(J_\perp E_{p\perp})$, can be instantaneously non-zero but averages to zero over a wave period[1]. In such an undamped wave, power sloshes back and forth between the wave field and particles with no net energy transfer. There are no corresponding fluctuations in $\Delta E_\parallel$ and $\Delta J_\parallel$ in an ideal Alfvén wave. For kinetic-scale Alfvén waves, however, non-zero $\Delta E_{p\parallel}$ fluctuations enable the Landau resonance, where particles with $V_\parallel \sim \omega / k_\parallel$ can gain or lose energy through interaction with the wave field. These interactions, combined with an imbalance in the number of particles that are moving faster than or slower than the wave, result in net plasma heating or cooling[4]. Here, fluctuations in $\Delta J_\parallel$ and $\Delta E_{p\parallel}$ become in-phase such that the wave-averaged $\Delta(J_\parallel E_{p\parallel})$ is non-zero[3,16]. Likewise, fluctuations in $\Delta B_\parallel$ result in transit-time damping effects, the magnetic analog of Landau damping, where the magnetic mirror force takes the place of $\mathbf{E}_p$[2,4]. For nonlinear KAWs, parallel fluctuations can be sufficiently large in amplitude to trap electrons between adjacent wave peaks. The oscillatory bounce motion of these electrons produces equal numbers of particles moving faster than or slower than the wave, limiting the effects of Landau and transit-time damping, and enabling stable wave mode propagation[4,17].

The detailed properties of KAWs (for example, $\Delta \mathbf{J}$, $\Delta \mathbf{E}_p$, $\mathbf{k}$) have been difficult to characterize due to their small spatial and temporal scales with respect to the capabilities of laboratory or on-orbit plasma instrumentation. Accurate estimates of current density and the characterization of particle populations require full three-dimensional distribution functions of both electron and ions on timescales faster than the wave frequency in the observation frame of reference. In addition, estimates of pressure gradients and wavevectors rely on multiple observation points being available within a single wave peak. However, NASA's recently launched Magnetospheric Multiscale (MMS) mission[18] consists of four identical observatories deployed in a tetrahedron configuration that measure charged particle and electromagnetic fields orders of magnitude more quickly than previous space

missions. This increased temporal sampling combined with a small MMS inter-spacecraft separation enables plasma parameters and their spatial gradients to be determined at kinetic scales.

Here we use observations from MMS to characterize the microphysics of a monochromatic Alfvén wave. Through the calculation of $\Delta \mathbf{J} \bullet \Delta \mathbf{E}$, we provide a direct measurement of the conservative energy exchange between the wave's electromagnetic fields and particles. A perpendicular spatial scale of $k_\perp \rho_i \sim 1$, non-zero $\Delta E_{p\parallel}$ and $\Delta J_\parallel$ fluctuations, and a parallel wave speed close to the local Alfvén speed confirm that the wave packet is an ion-scale KAW. Finally, analysis of the velocity distribution function of electrons reveals a population that is nonlinearly trapped within the wave's magnetic minima. These trapped electrons may have enabled nonlinear saturation of damping processes, resulting in marginally stable wave propagation and providing evidence in support of early analytical theories of wave–particle interactions in collisionless plasmas.

## Results

**Event overview.** On 30 December 2015, the four MMS observatories were near the dayside magnetopause, that is, the interface between the interplanetary magnetic field and the Earth's internal magnetic field, at [7.8, −6.9, 0.9] $R_e$ (1 $R_e = 1$ Earth radius = 6,730 km). Magnetic reconnection at the magnetopause boundary[19,20] generated a southward flowing exhaust at $\sim 22$:25 UT denoted by a $-V_z$ jet, an increase in plasma density, and a decrease in plasma temperature (see Fig. 1). There was no discernable rotation in the magnetic field suggesting that the spacecraft constellation remained inside the Earth's magnetosphere throughout this interval. Low frequency ($\sim 1$ Hz) waves were observed in the exhaust in a $\sim 4$ min interval localized to a region of strong proton temperature anisotropy ($T_{H+\perp}/T_{H+\parallel} \sim 2$). MMS partially crossed the magnetopause into the magnetosheath for the first time at $\sim 22$:35 UT (not shown) at [8.0, −6.9, 0.9] $R_e$. For the subsequent $\sim 2$ h, multiple magnetopause crossings resulted in the MMS spacecraft sampling both $+V_z$ and $-V_z$ jets, that is, above and below the reconnection site. However, $\sim 1$ Hz waves were only observed in the short interval shown in Fig. 1. The MMS observatories were in a tetrahedron configuration (quality factor[21] $\sim 0.9$) separated by $\sim 40$ km, a distance which corresponded to a local thermal ion gyroradius ($\rho_i = 35$ km).

The reconnection exhaust plasma consisted of mostly $H^+$ and some $He^{2+}$ with number density ratio $n_{He2+}/n_{H+} < 0.02$ throughout the interval. The local ratios of ion thermal parallel and perpendicular pressure to magnetic pressure were $\beta_\parallel \approx 0.2$ and $\beta_\perp \approx 0.5$, respectively. In addition, the average plasma flow velocity during this interval was $\mathbf{V}_o = [-17, 73, -183]$ km s$^{-1}$. This velocity corresponded to a jet flowing nearly anti-parallel to the background magnetic field ([0.10, −0.52, 0.85] direction) with speed $\sim 0.5\, V_A$, where $V_A$ is the Alfvén speed, that is, the characteristic speed in which information can be transferred along a magnetic field. For this interval, with $n_{H+} = 10$ cm$^{-3}$ and $B = 55$ nT, the local Alfvén speed was estimated to be 380 km s$^{-1}$. Variations were observed in the number density ($\Delta n$), bulk velocity ($\Delta \mathbf{v}_e$), temperature ($\Delta T_\parallel$, $\Delta T_\perp$) of both ions and electrons, and in the electric ($\Delta \mathbf{E}$) and magnetic fields ($\Delta \mathbf{B}$) (see Fig. 2). The amplitude of these $\sim 1$ Hz fluctuations were nonlinear with $\Delta n_{H+}/n_{H+} \sim 0.2$. The magnetic field fluctuations exhibited both left-handed and right-handed polarization (see Supplementary Fig. 1). Finally, bursts of electron phase space holes measured in the total parallel electric field ($\Delta E_\parallel$) were bunched with the wave in locations of strong electron pressure gradients.

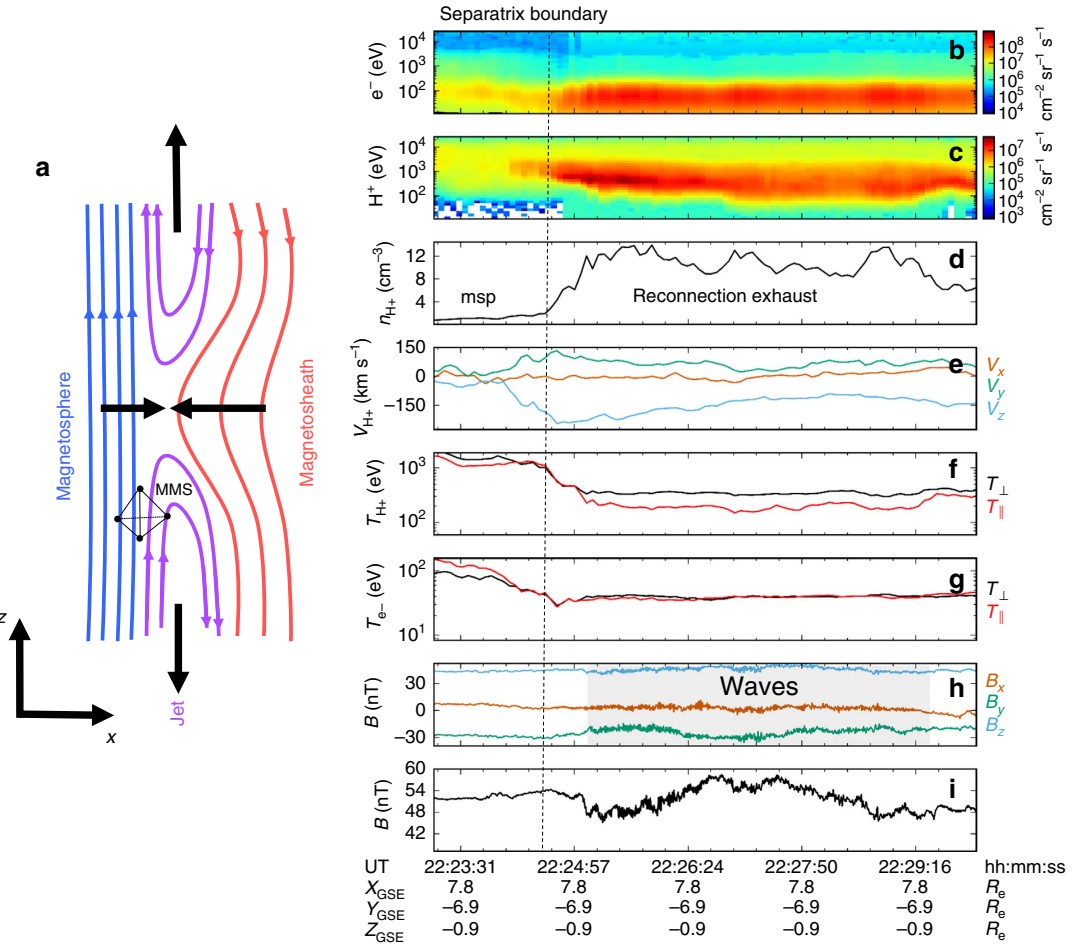

**Figure 1 | MMS observations of a reconnection exhaust. (a)** Illustration of the MMS constellation near the dayside magnetopause on 30 December 2015. MMS entered a southward flowing reconnection exhaust in the separatrix region on the magnetospheric (msp) side of the magnetopause. **(b–i)** Plasma parameters from MMS4 across the jet are shown from 22:23 to 22:30 UT. The density increased to ~10 cm$^{-3}$ **(d)** and $-V_z$ increased by ~200 km s$^{-1}$ **(e)**. No rotation in the magnetic field ($B$) indicated that the spacecraft remained inside the magnetosphere during this time period. Approximately 1 Hz waves **(h,i)** were observed to be localized in a region of enhanced ion temperature anisotropy, with $T_\perp/T_\parallel \sim 2$. H$^+$ dominated the ion composition during this time period.

**Wave properties**. Accurate determination of the wavevector (**k**) was critical to identify the observed wave mode. *In situ* estimation of **k**, especially for broadband wave spectra, is non-trivial and often relies on multi-spacecraft techniques[22]. Fortunately, the monochromatic nature of the observed wave enabled the application of several independent methods of wavevector determination. Here we utilized four methods to provide a robust estimate of **k**: (1) parallel component of the wavevector derived from the correlation between velocity and magnetic field fluctuations[16], (2) **k**-vector estimation from current and magnetic field fluctuations measured in the spacecraft frame[23,24], (3) comparison of spacecraft-measured gradients with their corresponding spacecraft-averaged quantities, that is, the plane-wave approximation[4], and (4) phase differencing of the magnetic field fluctuations between each spacecraft[25].

In the first method, we estimated the parallel component of the wavevector through comparison of four-spacecraft-averaged electron velocity and magnetic field fluctuations. Alfvén-branch waves have parallel wave speeds close to the local Alfvén speed, that is, $|\omega/k_\parallel| \approx V_A$ and correlated transverse fluctuations[16], $\Delta V_{e\perp} = -(\omega/k_\parallel)\Delta B_\perp/B$. Positively correlated ($R^2 = 0.92$) $\Delta V_{e\perp}$ and $\Delta B_\perp$ indicated that $\omega/k_\parallel = -1.15 \pm 0.03 \, V_A$, that is, the wave propagated anti-parallel to the background magnetic field near the Alfvén speed (see Supplementary Fig. 2). Although

qualitatively similar ~1 Hz fluctuations have been observed near Earth's bow shock that are more consistent with magnetosonic wave modes[26], a parallel phase speed well above the local sound speed of ~0.5 $V_A$ and the anti-correlation between density and magnetic field fluctuations were inconsistent with slow and fast magnetosonic wave modes, respectively.

In the second method, we combined fluctuations of current and magnetic field in the spacecraft frame to estimate **k** as a function of frequency using spectral techniques recently developed by Bellan[23,24]. Here the **k**-vector was derived directly from fluctuations in $\Delta \mathbf{J}$ and $\Delta \mathbf{B}$ measured in the spacecraft frame (see Fig. 3). Although this technique could have been applied to data from a single spacecraft, in order to maximize spectral resolution we used the four-spacecraft average of $\Delta \mathbf{B}$ and the average $\Delta \mathbf{J}$ determined from magnetometer data using the four-spacecraft 'curlometer' technique[27]. The value of **k** at the frequency of maximum spectral power, 0.9 Hz, was $\mathbf{k} = [7.1 \times 10^{-3}, \; -2.0 \times 10^{-2}, \; -2.2 \times 10^{-2}] \, \text{km}^{-1}$, which corresponded to a wavevector angle ($\theta$) of ~100° with respect to the background magnetic field and $k_\perp \rho_i \sim 1.0$.

In the third method, we used the phase difference[25] measured between each pair of MMS spacecraft for each component of the magnetic field to derive additional estimates of **k**. At the spectral peak of 0.9 Hz, the **k**-vector determined from the phase

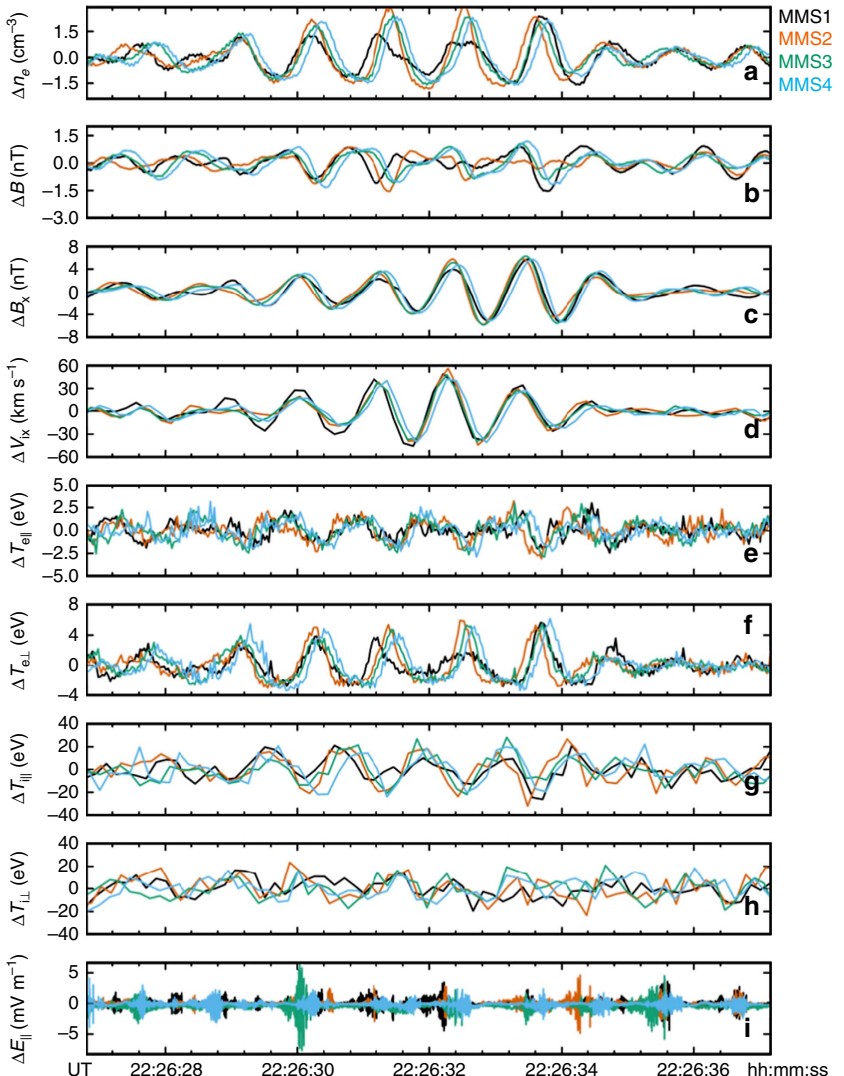

**Figure 2 | MMS observations of a KAW packet.** Plasma parameters measured by the four MMS observatories on 30 December 2015 in a KAW packet. (**a**,**b**) Compressive fluctuations are observed in anti-correlated electron density ($\Delta n_e$) and magnetic field magnitude ($\Delta B$) measurements. (**c**,**d**) Positively correlated fluctuations are observed in near-transverse components of the magnetic field ($\Delta B_X$) and electron bulk velocity ($\Delta V_{ex}$). (**e**–**h**) Fluctuations in both parallel and perpendicular temperature of both electrons ($\Delta T_e$) and ions ($\Delta T_i$) are shown, with the strongest relative fluctuations ($\sim 10\%$) observed in the perpendicular electron temperature. (**i**) Bursts of electron-scale phase space holes measured in the parallel electric field ($\Delta E_{||}$) are bunched with the ion-scale KAW wave and correspond to some of the gradients in the measured electron pressure.

differencing of the $B_X$, $B_Y$ and $B_Z$ fluctuations (using MMS3 as a reference) were: $[-7.4 \times 10^{-5}, -8.5 \times 10^{-3}, -1.5 \times 10^{-2}]$, $[2.9 \times 10^{-2}, 4.7 \times 10^{-3}, -1.1 \times 10^{-2}]$, and $[2.3 \times 10^{-2}, -3.5 \times 10^{-3}, -1.0 \times 10^{-2}]$ km$^{-1}$, respectively. Although similar phase shifts were observed in all components of $\Delta \boldsymbol{B}$ between MMS2, MMS3 and MMS4, there were significantly different shifts of MMS1 with respect to the other observatories for each component (see Supplementary Fig. 3). These differences demonstrated that this wave packet was not truly planar and exhibited spatial structure on the order of an ion gyroradius. Because MMS1 was farthest from the magnetopause (that is, the $X$ direction), the $k_X$ component was most strongly affected by this structure. Despite this discrepancy, all determinations of $\mathbf{k}$ result in $k_\perp \rho_i \sim 1$ and the phase differencing of $B_X$ and $B_Y$ components, those with the largest fluctuation power, both produced $\omega / k_{||} = -1.1\ V_A$.

Finally, in the fourth method, the small MMS spacecraft separations and high-quality tetrahedron formation enabled gradients of particle and field quantities to be estimated directly from the MMS data. These gradients were compared with those predicted by the plane-wave approximation (that is, '$\nabla \bullet$' $\approx \mathbf{i}\mathbf{k}$ and '$\nabla \times$' $\approx \mathbf{i}\mathbf{k} \times$ at a single frequency[4]) to both evaluate the validity of this approximation to the observed wave packet and to provide further validation of $\mathbf{k}$ (see Fig. 4). The current was calculated from three methods: (1) direct particle observations, that is, $e n_e (\mathbf{V}_i - \mathbf{V}_e)$, (2) magnetic field 'curlometer'[27], that is, $\nabla \times \mathbf{B}/\mu_0$, and (3) the plane-wave approximation, that is, $\mathbf{i}\mathbf{k} \times \mathbf{B}/\mu_0$. All three estimates of $\Delta \mathbf{J}$ are shown in Fig. 4. $k_y$ and $k_z$ most strongly influenced the plane-wave-derived currents such that this intercomparison was relatively insensitive to errors in the determination of $k_x$. The electron-pressure-gradient-driven electric field determined from four spacecraft measurements (that is, $-\nabla \bullet \underline{\mathbf{P}}_e/(n_e e)$), when compared with its plane-wave approximated value (that is, $-\mathbf{i}\mathbf{k} \bullet \underline{\mathbf{P}}_e/(n_e e)$), provides further confidence in the determination of $\mathbf{k}$ (see Fig. 4). Here all three components of $\mathbf{k}$ contributed to this result. The $X$-component comparison demonstrates that $k_x$ is of the correct sign but may underestimate the four-spacecraft gradient.

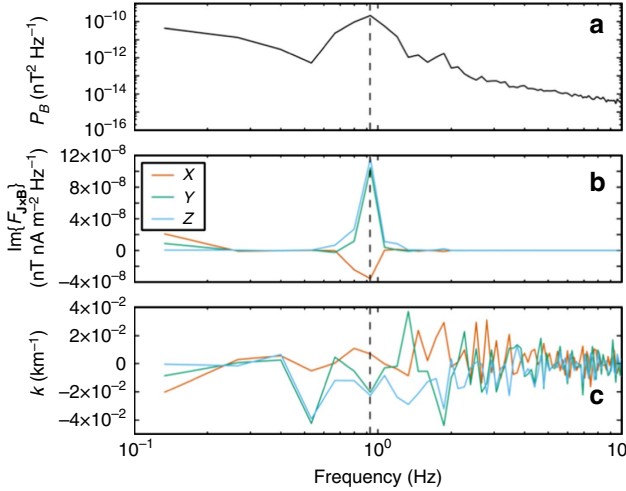

**Figure 3 | Wavevector estimated from current density fluctuations.**
(**a**) Power spectral density of MMS-averaged magnetic field magnitude from 22:26:28.18–22:26:35.83 UT, (**b**) the imaginary part of the Fourier amplitudes of fluctuations in MMS-averaged $\mathbf{J} \times \mathbf{B}$ and (**c**) corresponding components of $\mathbf{k}(\omega)$ derived using the Bellan[23,24] technique. At the spectral peak of $\sim 0.9$ Hz, $\mathbf{k} = [7.1 \times 10^{-3}, -2.0 \times 10^{-2}, -2.2 \times 10^{-2}]\,\mathrm{km}^{-1}$. This wavevector yielded $k_\perp \rho_i \sim 1$ and an angle of $\sim 100°$ with respect to the background magnetic field.

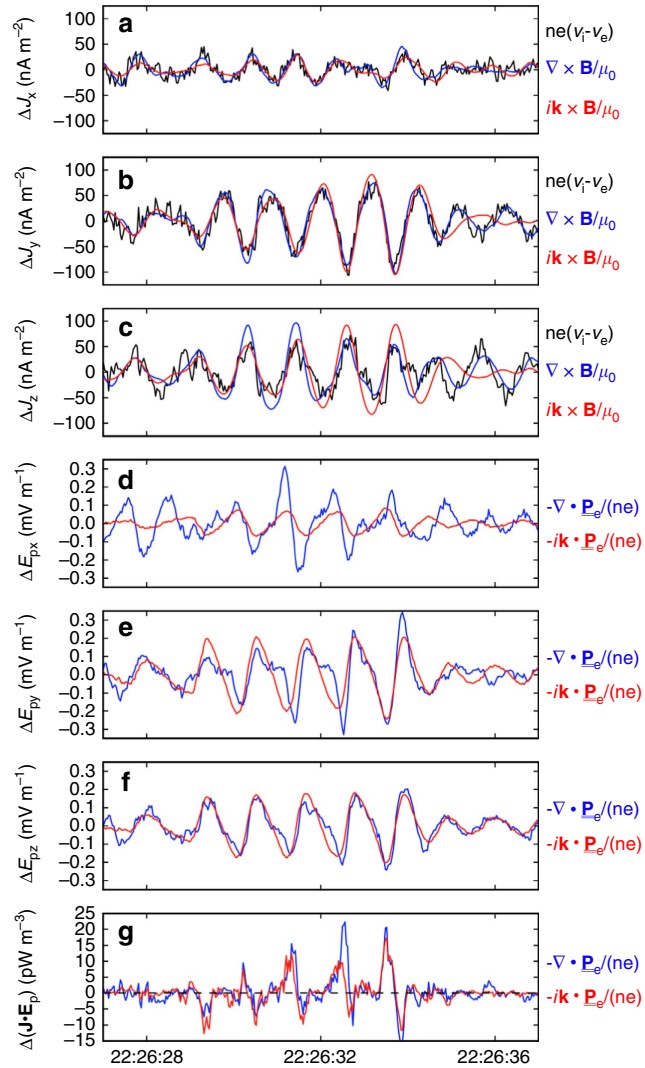

**Figure 4 | Comparison of current and electric field estimates.**
(**a–c**) MMS-averaged current fluctuations ($\Delta \mathbf{J}$) derived from the curlometer technique (blue), four-spacecraft-averaged particle observations (black) and four-spacecraft-averaged plane-wave approximation using $\mathbf{k} = [7.1 \times 10^{-3}, -2.0 \times 10^{-2}, -2.2 \times 10^{-2}]\,\mathrm{km}^{-1}$ (red). (**d–g**) MMS-averaged $\Delta \mathbf{E}_\mathrm{p}$ and $\Delta(\mathbf{J} \cdot \mathbf{E}_\mathrm{p})$ derived from the divergence of the electron pressure tensor (blue) and from the plane-wave approximation (red). Agreement between all quantities provides additional confidence in the estimation of $\mathbf{k}$.

We adopted the $\mathbf{k}$-vector derived using the Bellan[23,24] method $\mathbf{k} = [7.1 \times 10^{-3}, -2.0 \times 10^{-2}, -2.2 \times 10^{-2}]\,\mathrm{km}^{-1}$ because it simultaneously leveraged data from all four spacecraft and all components of the magnetic field. Allowing for $\sim 30\%$ (3-$\sigma$ level) uncertainty in each individual component, we found $k_\perp \rho_i = 1.02 \pm 0.07$ with wavevector angle $104 \pm 4°$ from the magnetic field. The 0.9 Hz peak observed in the spacecraft frame ($\omega_\mathrm{sc}$) was then Doppler-shifted by $\omega = \omega_\mathrm{sc} - \mathbf{k} \cdot \mathbf{V}_\mathbf{o}$ to obtain a frequency of $\omega/\omega_\mathrm{ci,H+} = 0.61 \pm 0.08$ in the plasma frame. We conclude that multiple independent methods indicated that MMS resolved a kinetic-scale Alfvén-branch wave.

**Modelled wave growth rates.** Growth rates ($\gamma = \mathrm{Im}\{\omega/\omega_\mathrm{ci}\}$) and polarization ($\mathrm{Re}\{iE_y/E_x\}$) solutions along the Alfvén-branch dispersive surface were estimated using a linear dispersion solver and are shown as a function of $\theta$ in Fig. 5. The dispersion solver predicted that the large ion temperature anisotropy of $T_{\mathrm{i}\perp}/T_{\mathrm{i}\parallel} \sim 2$ produced a nearly monochromatic ion cyclotron wave mode that propagated parallel/anti-parallel to the background magnetic field ($\theta = 0°, 180°$) with $\omega/\omega_\mathrm{ci} \sim 0.5$, $k\rho_i \sim 0.4$ and left-handed polarization. At increasingly oblique wavevector angles, the predicted wave growth was substantially reduced. There was no slow or fast magnetosonic wave growth predicted for the measured plasma parameters. Several Alfvén-branch dispersion curves are shown in Fig. 5 as a function of $k\rho_i$ and $\theta$. The observed KAW mode ($\omega/\omega_\mathrm{ci} = 0.6$, $k\rho_i = 1$, $\theta = 100°$) was close to but not precisely on the solution surface. Nearby Alfvénic solutions to the measured data (matching two of the three wave parameters) were $\{\omega/\omega_\mathrm{ci} = 0.3, k\rho_i = 1, \theta = 100°\}$, $\{\omega/\omega_\mathrm{ci} = 0.6, k\rho_i = 1.6, \theta = 100°\}$ and $\{\omega/\omega_\mathrm{ci} = 0.6, k\rho_i = 1, \theta = 110°\}$. All of these nearby solutions were weakly damped ($|\gamma| \sim 10^{-2}$) such that local generation of the observed KAW was not predicted by linear wave theory. However, local spatial gradients of plasma density may have increased the $\theta$ of the ion cyclotron mode during its propagation, converting it into an oblique Alfvén wave[4]. Furthermore, nonlinear effects and parametric forcing (for example,

magnetopause motion) were not taken into account by the homogenous dispersion solver, yet may have played a role in the evolution of the observed KAW.

**Wave–particle interactions.** Given the demonstrated validity of the plane-wave approximation for $\Delta \mathbf{E}_\mathrm{p}$, the electron-pressure-gradient-driven electric field was estimated at a single spacecraft, for example, MMS4, using $-i\mathbf{k} \cdot \underline{\mathbf{P}}_\mathbf{e}/(n_e e)$. Fluctuations of $\Delta \mathbf{E}_\mathrm{p}$ and $\Delta \mathbf{J}$ in magnetic coordinates on MMS4 are shown in Fig. 6. In addition to the transverse electric field fluctuations expected for all Alfvén waves, fluctuations in $\Delta E_{\mathrm{p}\parallel}$ further confirmed the presence of kinetic-scale effects. These parallel fluctuations were an order of magnitude smaller than those in $\Delta E_{\mathrm{p}\perp}$ as expected from KAW theory[16]. Furthermore, fluctuations in all components of $\Delta \mathbf{J}$ and $\Delta \mathbf{E}_\mathrm{p}$ (both perpendicular and parallel) were each

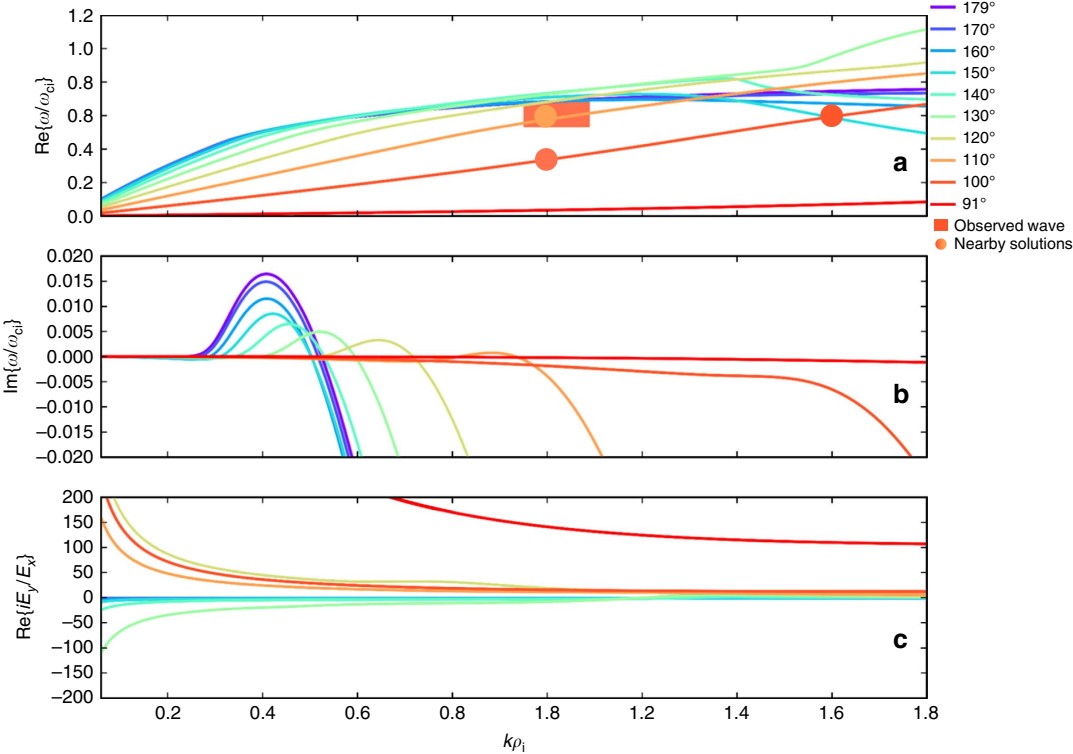

**Figure 5 | Modelled dispersion curves for the local plasma environment.** (**a**) The real part of $\omega/\omega_{ci}$, that is, the wave oscillation frequency, (**b**) the imaginary part of $\omega/\omega_{ci}$, that is, the wave growth/damping rate and (**c**) the real part of $iE_x/E_y$, that is, the polarization of the wave, as a function of scaled wavevector magnitude $k\rho_i$. Coloured curves correspond to solutions of a linear dispersion relation solver taken along the Alfvén branch for different wavevector angles ($\theta$) relative to the background magnetic field. The fastest growing wave mode has a wavevector parallel/anti-parallel to the background magnetic field (that is, $\theta = 0°$, 180°) at $\omega/\omega_{ci} \sim 0.5$ and $k\rho_i \sim 0.4$ and is left-hand polarized (that is, Re{$iE_x/E_y$} < 0). A transition to right-hand polarization (that is, Re{$iE_x/E_y$} > 0) occurred at $\theta \sim 130°$. No strong growth or damping was predicted for the observed KAW ($\theta = 104 \pm 4°$, $\omega/\omega_{ci} = 0.61 \pm 0.08$ and $k_\perp \rho_i = 1.02 \pm 0.07$), indicated with the shaded area in (**a**). The dimensions and colour of the shaded area correspond to the reported uncertainties of the measured $\omega/\omega_{ci}$ and $k_\perp \rho_i$ parameters and $\theta \approx 100°$, respectively Nearby solutions that match two of the measured {$\omega/\omega_{ci}$, $k\rho_i$, $\theta$} parameters (but not all three) are shown as solid circles. The colour of each circle corresponds to the wavevector angle.

$\sim 90°$ out of phase with one another. These phase differences resulted in a non-zero instantaneous value of $\Delta(\mathbf{J} \bullet \mathbf{E_p})$ with $\Delta |\mathbf{J} \bullet \mathbf{E_p}|_{max} \approx 50 \text{ pW m}^{-3}$ and near-zero wave-averaged $\Delta(J_\perp E_{p\perp})$ and $\Delta(J_{||} E_{p||})$ quantities. These data demonstrated the conservative energy exchange between the particles and fields that comprise an undamped KAW.

Because $k_\perp \rho_e \ll 1$, electrons should have remained magnetized throughout the wave packet. Close examination of the electron velocity distribution function in the parallel wave frame revealed three distinct populations of electrons in the wave packet: (1) an isotropic thermal core, (2) suprathermal beams counterstreaming along the magnetic field, and (3) trapped particles with near $\sim 90°$ magnetic pitch angles (Fig. 7). Thermal and counter-streaming electrons are commonly observed in the magnetopause boundary layer in the absence of analogous wave activity[28]. However, trapped electron distributions are atypical of ambient boundary layer plasmas. Furthermore, these trapped electrons were dynamically significant: they accounted for $\sim 50\%$ of the density fluctuations within the KAW. Although these electrons also resulted in a $\sim 20\%$ increase in $T_{e\perp}$, they were not indicative of heating but rather of a nonlinear capture process.

The depth of the parallel potential well estimated from $\Delta E_{p||}$ and $k_{||}$ was found to be $\sim 10 \text{ V}$ (Fig. 7). In addition, the parallel magnetic field of the wave generated a mirror force that resulted in a kinetic-scale magnetic bottle between successive wave peaks. This mirror force supplemented the force from the wave's parallel electric field, enabling trapping of electrons with magnetic pitch

angles between $\sim 75°$ and $\sim 105°$ ($B_{min}/B_{max} = 0.96$). To understand the combined effects of these forces, electrons measured in the magnetic minima were Liouville-mapped to other locations along the wave using various parallel potential well depths (Fig. 8). The full-width at half maximum distance along the wave at a pitch angle of 90° was calculated for each potential and compared with the measured data. The best match between measured and Liouville-mapped distributions was found for a potential well depth of $|\Phi_{max}| = 10 \text{ V}$. Such agreement provided additional validation of $\Delta E_{p||}$ and $k_{||}$. In addition, these distributions demonstrated that the effect of the parallel electric field was to confine magnetically trapped electrons closer to magnetic minima.

## Discussion

KAWs in turbulent space plasmas are thought to account for heating of plasmas at kinetic scales[5-7]. In previous studies[29,30], such waves were found to have $k_\perp \gg k_{||}$, that is, $\theta \sim 90°$. This plasma heating was accompanied by significant reductions in field fluctuation power. The wave presented here had a somewhat higher frequency ($\omega_{ci,He2+} < \omega < \omega_{ci,H+}$) than those considered in these previous KAW studies ($\omega \ll \omega_{ci,H+}$, $\omega_{ci,He2+}$). Furthermore, its comparatively non-perpendicular wavevector ($\theta \approx 100°$) and large scale ($k_\perp \rho_i \approx 1$) indicated that the observed wave was close to the transition point between ideal and kinetic regimes. Nonetheless, the wave had non-zero $\Delta J_{||}$ and $\Delta E_{p||}$

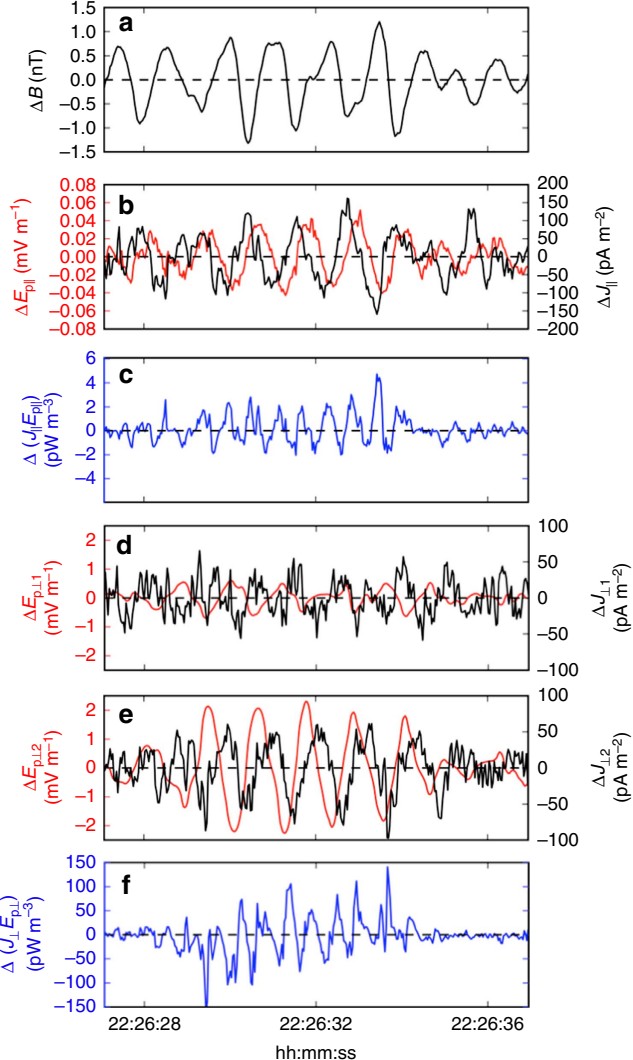

**Figure 6 | Current and electric field fluctuations in a KAW.** Fluctuations in (**a**) magnetic field magnitude $\Delta B$, (**b**) parallel electric field $\Delta E_{p\|}$ and parallel current $\Delta J_{\|}$, (**c**) $\Delta(J_{\|}E_{p\|})$, (**d**,**e**) perpendicular electric fields ($\Delta E_{p\perp 1}$ and $\Delta E_{p\perp 2}$) and current ($\Delta J_{\perp 1}$ and $\Delta J_{\perp 2}$) and (**f**) $\Delta(J_{\perp}E_{p\perp})$ observed by MMS4 on 30 December 2015 between 22:26:27 and 22:26:37 UT. Pressure-gradient-driven electric field quantities were inferred from the **k**-vector and electron pressure tensor from MMS4 using the plane-wave approximation (that is, $\mathbf{E}_p = -i\mathbf{k}\bullet\underline{\mathbf{P}}_e/n_e e$). Current densities were derived directly from MMS4 particle observations. Current density and electric field fluctuations were 90° out of phase in both the perpendicular and parallel directions, resulting in non-zero instantaneous $\Delta(\mathbf{J}\bullet\mathbf{E}_p)$, which provided confirmation of the conservative energy exchange between the wave field and plasma particles. The amplitude of $\Delta(J_{\perp}E_{p\perp})$ was an order of magnitude higher than $\Delta(J_{\|}E_{p\|})$. The wave-averaged $\Delta(\mathbf{J}\bullet\mathbf{E}_p)$ was approximately zero, indicating that the wave was in a marginally stable state, that is, was neither growing nor damping. Quantities are shown in magnetic coordinates.

fluctuations, confirming that it contained kinetic-scale structure not present in an ideal Alfvén wave. These observations demonstrated that the mere presence of a KAW or parallel electric field fluctuations do not necessarily imply heating via Landau damping. Only in-phase fluctuations in $\Delta\mathbf{J}$ and $\Delta\mathbf{E}_p$ result in such net transfer of energy from the wave field to the plasma particles.

In linear KAW theory, the electrostatic field formed by parallel gradients in electron pressure enables the energization

of particles via the Landau resonance[4,13,16]. Similarly, the transit-time resonance becomes relevant for systems where there are parallel gradients in magnetic field magnitude. Despite the presence of these field gradients in the observed KAW, out-of-phase $\Delta E_{p\|}$ and $\Delta J_{\|}$ fluctuations and a finite wave amplitude for several wave periods (that is, $|\gamma| \ll 1$) indicated the absence of strong wave growth or damping. Although a hot core population ($V_{th,e} \gg |\omega/k_{\|}|$) does not lead to strong damping (Fig. 5), the velocity distribution function of electrons was not directly sampled at energies corresponding to $V_{\|} \sim \omega/k_{\|}$ (that is, ~0.5 eV). Electrons at these low energies are often present as they serve to neutralize a ubiquitous population of 'hidden' cold ions that flow out from the ionosphere[31]. Such ionospheric electrons may have added structure to the velocity distribution function near $V_{\|} \sim \omega/k_{\|}$, amplifying damping rates. However, nonlinear KAW theories have predicted that trapped electrons with $V_{\|} \sim \omega/k_{\|}$ lead to wave stabilization if their bounce frequency ($\omega_B$) is significantly faster than the damping or growth rate, that is, $\omega_B/\omega_{ci} \gg |\gamma|$[4,17,32]. We estimated $\omega_B/\omega_{ci} \sim 1$ for this wave, consistent with such a criterion. Therefore, the presence of trapped electrons here could have contributed to nonlinear instability saturation in a single-mode wave even if there were low energy structure in the electron distribution function that was not resolved by MMS.

Finally, at higher frequencies (~1 kHz), fluctuations in the total parallel electric field $\Delta E_{\|}$ associated with electron phase space holes[33] were bunched in phase with the low frequency wave packet (Fig. 1). Because these structures persisted outside of the KAW interval (not shown), it is unlikely that they were related to its initial generation. However, the location of these electron-scale structures within the wave was coincident with the location of electron pressure gradients, suggesting that they could have contributed, in an average sense, to some of the observed ion-scale $\Delta E_{p\|}$ fluctuations. Furthermore, electron holes may have been responsible for higher frequency contributions to $\Delta(J_{\|}E_{\|})$ in the form of nonlinear and turbulent terms in the electron momentum equation[34].

Using MMS data, we have experimentally confirmed the conservative energy exchange between an undamped kinetic Alfvén wave field and plasma particles: fluctuations of all three components of $\Delta\mathbf{J}$ and $\Delta\mathbf{E}_p$ were 90° out of phase with one another, leading to instantaneous non-zero $\Delta(\mathbf{J}\bullet\mathbf{E}_p)$. Furthermore, we have discovered a significant population of electrons trapped within adjacent wave peaks by the combined effects of the parallel electron-pressure-gradient-driven electric field and the magnetic mirror force. In addition to contributing ~50% of the density fluctuations in the wave, these trapped electrons may have provided nonlinear saturation of Landau and transit-time damping. The monochromatic nature of the wave enabled a direct comparison of observations with linear and nonlinear KAW theories. It is crucial to understand these dynamics to predict the evolution of kinetic-scale waves in laboratory fusion reactors, planetary magnetospheres and astrophysical plasmas.

## Methods

**Coordinate systems.** The coordinate system used in this study (unless otherwise noted) was the Geocentric Solar Ecliptic (GSE) coordinate system, where the $X$ direction pointed towards the Sun along the Earth–Sun line, the $Z$ direction was oriented along the ecliptic north pole and the $Y$ direction completed the right-handed coordinate system[35]. Local 'magnetic coordinates' were derived from GSE vectors where $\mathbf{B}_3$ was parallel to the local magnetic field direction, $\mathbf{B}_1$ was in the $\mathbf{X}_{GSE} \times \mathbf{B}_3$ direction and $\mathbf{B}_2$ completed the right-handed coordinate system, that is, $\mathbf{B}_1 \times \mathbf{B}_2 = \mathbf{B}_3$.

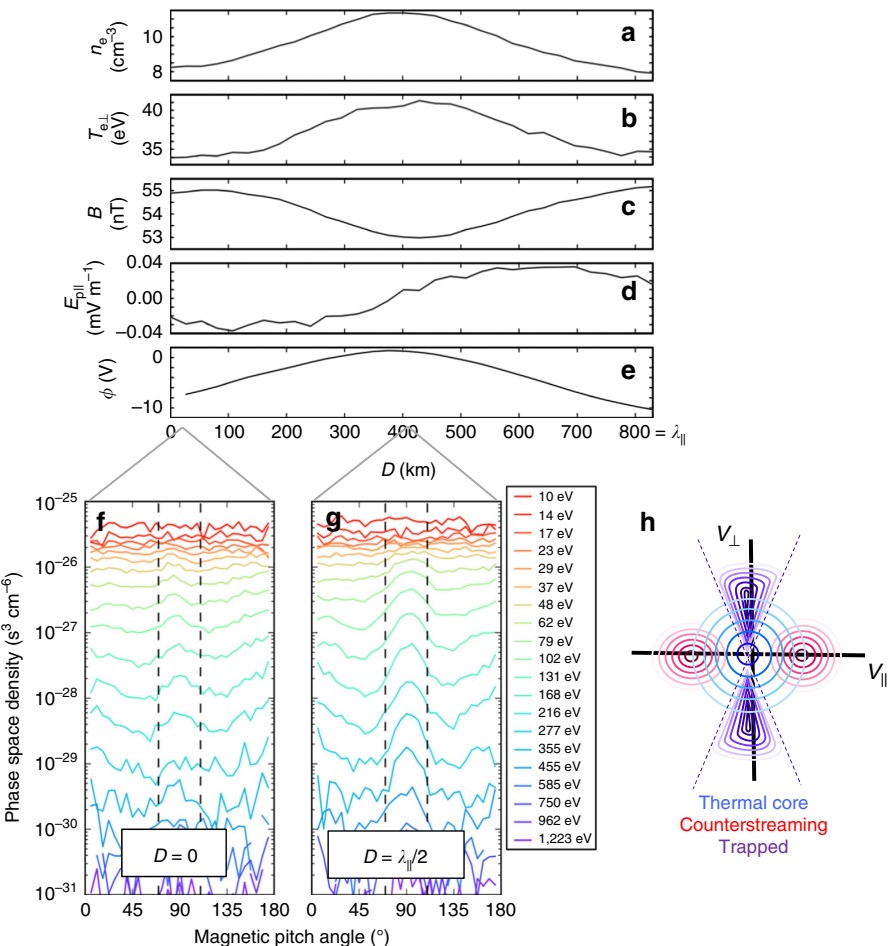

**Figure 7 | Structure inside of a KAW packet.** Profile of (**a**) density $n_e$, (**b**) perpendicular electron temperature $T_{e\perp}$, (**c**) magnetic field magnitude $B$, (**d**) parallel electric field $\Delta E_{p\parallel}$ inferred from electron pressure gradients and (**e**) parallel potential $\Phi$ integrated from $\Delta E_{p\parallel}$ as a function of position $D$ in the wave for MMS4 from 22:26:29.94 to 22:26:30.90 UT. The reference value for the potential ($\Phi = 0$) was taken at the centre of the wave, that is, at the magnetic minimum. The wave had a parallel wavelength of $\lambda_\parallel \sim 830$ km or $\sim 20~\rho_i$. The ratio of the minimum to maximum magnetic field magnitude was $B_{min}/B_{max} = 0.96$, which was sufficient to trap electrons with magnetic pitch angles between $\sim 75°$ and $\sim 105°$. Phase space density as a function of energy and magnetic pitch angle are shown at the magnetic (**f**) maximum ($D = 0$) and (**g**) at the magnetic minimum ($D = \lambda_\parallel/2$) in the wave frame of reference (that is, all measured velocities shifted by $-V_A$ along the magnetic field direction). An illustration of three corresponding populations of electrons is shown in $V_\parallel - V_\perp$ space in panel (**h**). Thermal (energies below $T_e \approx 35$ eV) electrons have nearly isotropic pitch-angle distributions (blue contours). Suprathermal (energies above $T_e$) electrons were observed as peaks in the phase space density at pitch angles near 0° and 180° (red contours). Finally, a trapped population with energies above $T_e$ is shown between the dashed vertical lines (purple contours). These trapped electrons were responsible for the increased perpendicular temperature at the magnetic minima and accounted for $\sim 50\%$ of the increase in density.

**Calculation of plasma parameters.** The thermal gyroradius was calculated using

$$\rho_i = \frac{m_{H+} \sqrt{\frac{k_B T_{H+\perp}}{m_{H+}}}}{eB} \qquad (1)$$

where $k_B$ is Boltzmann's constant, $e$ is the elementary charge and $m_{H+}$ is the mass of $H^+$. The ion gyrofrequency was calculated using,

$$\omega_{ci} = \frac{eB}{m_{H+}} \qquad (2)$$

The plasma thermal pressure was calculated using $n_{H+} k_B T_{H+}$. The magnetic pressure was calculated using $B^2/2\mu_o$ where $\mu_o$ is the magnetic permeability of free space. Finally, the Alfvén speed was calculated using

$$V_A = \frac{B}{\sqrt{\mu_o n_{H+} m_{H+}}} \qquad (3)$$

All calculations were done in SI units.

**$\Delta V_e$–$\Delta B$ correlations.** The comparison of $\Delta V_e$ and $\Delta B$ was done in the direction of minimum current density fluctuations ([0.93, 0.32, 0.18]) such that ion and electron velocities were approximately equal. This minimum variance direction

was nearly perpendicular to the background magnetic field direction $\mathbf{b} = [0.10, -0.52, 0.85]$.

**Electric field measurements.** The electric field in the electron frame was defined as $\mathbf{E} + \mathbf{V}_e \times \mathbf{B}$, where $\mathbf{E}$ was the measured electric field in the spacecraft frame[23]. Since $J$ is frame independent, this electron-frame electric field is conveniently used for estimates of energy transfer, that is, plasma heating occurs when $\mathbf{J} \bullet (\mathbf{E} + \mathbf{V}_e \times \mathbf{B}) > 0$. At the scales relevant for this KAW packet, electrons remained magnetized such that electron inertia and anomalous resistivity contributions to the electric field were neglected and the pressure gradient term should have been the dominant contributor to $\mathbf{E} + \mathbf{V}_e \times \mathbf{B}$ at low frequencies. The individual amplitudes of $\mathbf{E}$ and $\mathbf{V}_e \times \mathbf{B}$ were measured to be on the order of several mV m$^{-1}$. Systematic uncertainty in both particle and fields measurements would have led to a challenging recovery of $\mathbf{E} + \mathbf{V}_e \times \mathbf{B}$ because $|\mathbf{E} + \mathbf{V}_e \times \mathbf{B}| \ll |\mathbf{E}|, |\mathbf{V}_e \times \mathbf{B}|$. Therefore, accurate direct estimates of $\mathbf{J} \bullet (\mathbf{E} + \mathbf{V}_e \times \mathbf{B})$ were not recovered for this event. Instead, here we focussed on effects of the electric field generated by the divergence of the electron pressure tensor, that is, $\mathbf{E}_p = -\nabla \bullet \underline{\underline{\mathbf{P}}}_e/(n_e e)$ and validated the measurement using multiple methods. In the electron frame, the electrons are not moving so there is no magnetic term in the electron equation of motion giving $\mathbf{E} \approx \mathbf{E}_p$.

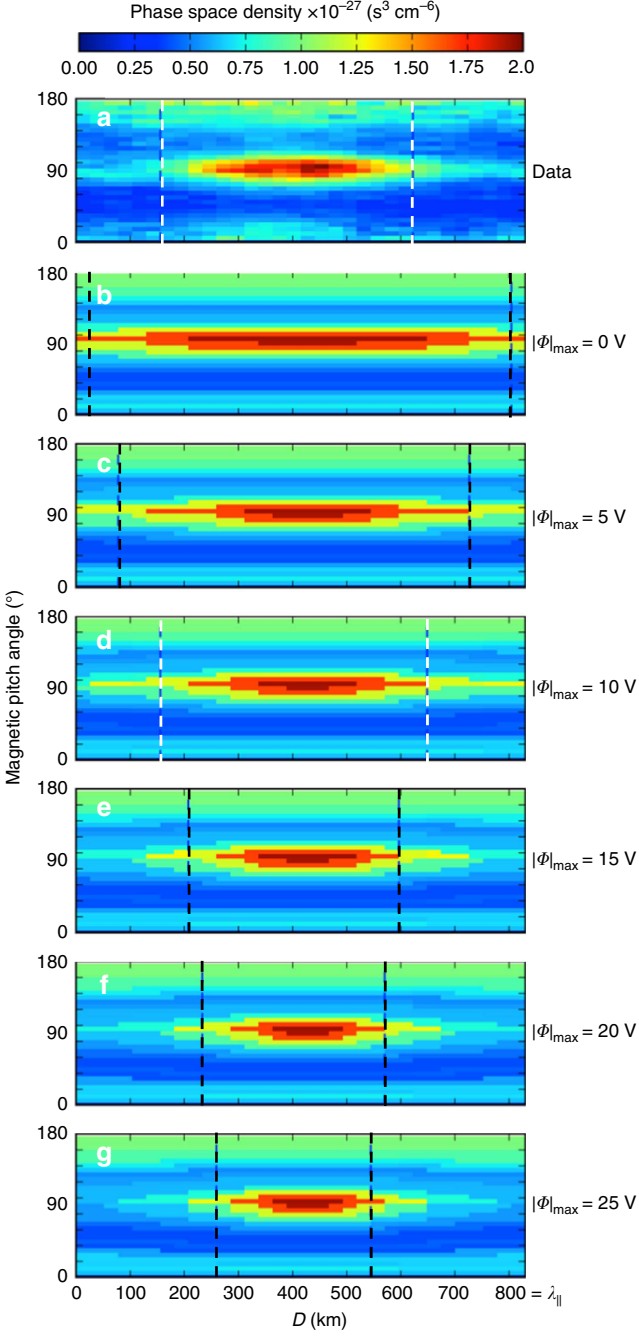

**Figure 8 | Liouville-mapped electrons in a KAW.** Measured phase space densities from MMS4 as a function of magnetic pitch angle and position in the wave, $D$, between successive magnetic field maxima in the KAW packet from Fig. 3 (22:26:29.94–22:26:30.90 UT) for 132 eV electrons. Liouville-mapped distributions are shown for $|\Phi|_{max} = 0$, 5, 10, 15, 20 and 25 V (**a–g**). These distributions were constructed using measured phase space densities at the magnetic minimum (that is, $D = \lambda_{||}/2$). The mirror ratio of $B_{min}/B_{max} = 0.96$ confined particles to pitch angles between 75° and 105° in all cases. The parallel potential formed from $\Delta E_{p||}$ provided additional spatial localization of the trapped population within the wave minima. Vertical dashed lines denote the full-width at half-maximum along $D$ at a pitch angle of 90°. The best agreement with the measured data occurred for the distribution mapped using $|\Phi|_{max} = 10$ V, which was consistent with independent estimates of $k_{||}$ and $\Delta E_{p||}$.

**Linear instability analysis.** To determine the properties of kinetic modes that interact with ions and electrons at their respective scales, we used the linear dispersion solver PLADAWAN[36] (PLAsma Dispersion And Wave ANalyzer) to solve the linearized Vlasov-Maxwell system for arbitrary wavevector directions. Using measured plasma parameters of ions and electrons, the dispersion solver produced growth rates and wave properties as functions of $\omega$ and **k**. The plasma parameters used as input to the dispersion solver (assuming stationary plasma) were $n_{e-} = 10$ cm$^{-3}$, $B = 55$ nT, $T_{e\perp} = T_{e||} = 35$ eV, $T_{H+||} = 175$ eV and $T_{H+\perp} = 350$ eV. Wave polarization was calculated using the simulated electric field fluctuations as Re$\{iE_x/E_y\}$. Left-hand and right-hand polarization corresponded to Re$\{iE_x/E_y\} < 0$ and Re$\{iE_x/E_y\} > 0$, respectively[4]. No growth was observed for the slow-mode or fast-mode magnetosonic branches of the dispersion relation. Additional simulations were run to evaluate the influence of He$^{2+}$ on the observed instability. Increased $n_{He2+}/n_{H+}$ ratios up to 0.02 with $T_{He2+} = 550$ eV reduced the maximum wave growth but did not alter the sharpness of the peak in $k$-space. No new wave modes appeared to be introduced into the system from the presence of the local He$^{2+}$ population.

**Liouville mapping and electron bounce motion.** Under the assumption that electron phase space density $f(\mathbf{v})$ was conserved along particle trajectories throughout the wave interval (that is, Liouville's theorem), we used $f(\mathbf{v})$ measured in the magnetic minimum, defined as $f_o(\mathbf{v})$, a sinusoidal profile of the magnetic field strength $B$ with $M = B_{min}/B_{max} = 0.96$, and a sinusoidal profile of electric potential $\Phi$ to infer the velocity distribution along the wave[37,38]. Velocity space was transformed using equations

$$v_{||o} = \pm \sqrt{v_\perp^2(D)\left(1 - \frac{B_o}{B(D)}\right) + v_{||}^2(D) - \frac{2e}{m_e}\Phi(D)} \quad (4)$$

and

$$v_{\perp o} = \sqrt{v_\perp^2(D)\left(\frac{B_o}{B(D)}\right)}, \quad (5)$$

where the 'o' subscripts denote values at the magnetic minimum of the wave. The '+' and '−' branches of equation (4) correspond to the sign of $v_{||}$. For each $(v_{||}, v_\perp)$ point in the reconstructed skymap, equations (4 and 5) provided a point $(v_{||o}, v_{\perp o})$ that was used to map a phase space density in the reference distribution, that is, $f(v_{||}, v_\perp) = f_o(v_{||o}, v_{\perp o})$.

In the magnetic minimum ($D = \lambda_{||}/2$), $\frac{B_o}{B(D)} = 1$ and $\Phi = \Phi_o = 0$. At the magnetic maximum ($D = 0, \lambda_{||}$), $\frac{B_o}{B(D)} = M$ and $\Phi = -|\Phi_{max}|$, that is,

$$\frac{B_o}{B(D)} = M + (1 - M)\sin\left(\frac{\pi}{\lambda_{||}}D\right) \quad (6)$$

$$\Phi(D) = -\frac{|\Phi_{max}|}{2}\left(1 + \cos\left(\frac{2\pi}{\lambda_{||}}D\right)\right). \quad (7)$$

Finally, bounce frequencies ($\omega_B = 1/\tau_B$) for trapped electrons were estimated using

$$\tau_B = 4\int_{\lambda_{||}/2}^{R}\frac{dD}{v_{||}(D)}, \quad (8)$$

where $R$ was defined as the reflection point along the wave (that is, $v_{||}(R) = 0$). Electrons with pitch angles 75–90° and energies 100–400 eV produced bounce frequencies of $1.4 \pm 0.3$ Hz (that is, $\omega/\omega_{ci} = 1.6 \pm 0.3$) in a $\lambda_{||} = 830$ km wave with $M = 0.96$.

**MMS data sources and processing.** Particle, magnetic field and electric field data were measured by the Fast Plasma Investigation[39] (FPI), the Fluxgate Magnetometers[40] and Electric Field Double Probe[41] instruments, respectively. Corresponding composition data at ~10 s time resolution was obtained from the Hot Plasma Composition Analyzer[42]. Time series data were high-pass filtered with a fifth-order digital Butterworth IIR filter with coefficients $b = [0.85850229, -4.29251147, 8.58502295, -8.58502295, 4.29251147, -0.85850229]$ and $a = [1.0, -4.69504063, 8.82614592, -8.30396669, 3.90989399, -0.73702619]$, where $b$ and $a$ correspond to the filter's numerator and denominator polynomials listed in increasing order. This filter had an effective cutoff frequency of 0.5 Hz and no discernable effect (<1%) on the amplitude or phase of a 0.9 Hz input signal.

**Data availability.** Data used for this study is available to download from the MMS Science Data Center (https://lasp.colorado.edu/mms/sdc/) or from the corresponding author upon request.

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

## Acknowledgements

We thank the members of the FPI ground operations and science team for their feedback and support and Lynn Wilson for his insights about wave properties. This research was supported by the NASA Magnetospheric Multiscale Mission in association with NASA contract NNG04EB99C and by NSF award 1059519, AFOSR award FA9550-11-1-0184 and DOE awards DE-FG02-04ER54755 and DE-SC0010471. S.J.S. is grateful to the Leverhulme Trust for its award of a Research Fellowship.

## Author contributions

D.J.G. conducted the majority of the scientific and data analysis and was responsible for initial preparation of the manuscript text. A.F-V. assisted with the interpretation of wave signatures, plasma wave modeling and with the preparation of the manuscript text. J.C.D., S.A.B. and L.A.A. assisted with the interpretation of plasma wave signatures, detailed analysis of plasma data and with the preparation of the manuscript text. P.M.B. assisted with the implementation of the wavevector determination method and with the preparation of the manuscript text. S.J.S. assisted with the Liouville mapping of electron data and preparation of the manuscript text. B.L., V.N.C., M.O.C., Y.S. and W.R.P. provided and ensured quality of high-resolution plasma data and assisted with the preparation of the manuscript text. S.A.F. provided and ensured the quality of the plasma composition data. R.E.E. and R.B.T. provided and ensured the quality of high-resolution electric field data and assisted with the preparation of the manuscript text. R.J.S. and C.T.R. provided and ensured the quality of high-resolution fluxgate magnetometer data. B.L.G., C.J.P. and J.L.B. provided institutional and mission-level support of the analysis and ensured overall quality of MMS and FPI data.

## Additional information

**Competing interests:** The authors declare no competing financial interests.

**Publisher's note**: 

