## [Peer Review File · Nature Communications]

Reviewer #1 (Remarks to the Author):

Review of Gershman et al.

I have relatively little to say about this paper.

It is one of the first results coming from the MMS mission, highlighting the extraordinarily detailed way in which various plasma physics phenomena can now be measured in space. It is especially nice and valuable that many different fields simultaneously (magnetic, electric, density) can be measured and resolved on Larmor scales.

The observation of a KAW packet presented by the authors is an illustration of these great new instrumental capabilities. They are able to measure local $j \cdot E$ heating, which is indeed a first and so a milestone. I found the descriptions in the MS convincing and clear. It is interesting (although perhaps not entirely surprising) that the KAW traps some particles and it is impressive that this trapped population can be measured and diagnosed. This is space plasma being used as a plasma laboratory par excellence, something to be admired and encouraged.

As far as the physics is concerned, I must confess that the paper has left me cold. So, a KAW wave packet, so it damps on particles, so what? Nice addition to a "stamp collection" of space phenomena, but there is nothing here that would tell me what this is actually good for. Is this generic? Is this universal? Is this at least a triumphant observational verification of some beautiful piece of analytical theory? (O'Neil 1965 repeated referred to here is indeed an analytical tour de force, but it has little to do with KAWs specifically.)

Thus, I am excited that things like this can now be done, as far as instrumentation/technique are concerned, but regarding the actual physics result, I shrug my shoulders and get on with my life.

Do I think this should be published? Absolutely. Is it appropriate for Nature Comms? I think so, yes. Authors pay a hefty fee for Nature Comms, so the journal is designed to promote work involving big money, aka "big science". This work is indeed genuine big science — a lot of effort, expertise and money does go into being able to achieve technically accomplished feats of observation like this. This is a good thing too, as we will eventually learn new and exciting physics from MMS. It is only appropriate that an expensive journal like Nature Comms should promote expensive new capabilities like this.

Reviewer #2 (Remarks to the Author):

Review of

Wave-particle energy exchange directly observed in a kinetic Alfvén-branch wave

by Gershman et al.

Summary:

This is a timely paper that describes a detailed look at the dynamics of kinetic Alfvén waves (KAWs) using high quality, direct measurements from the MMS mission. The paper uses several independent, cutting-edge analysis techniques to determine the wavevector of the monochromatic KAW observed in the measurements. Overall, the paper presents a nice,

direct measurement of the conservative energy transfer between the fields and particles (as determined by computing $\mathbf{J} \cdot \mathbf{E}$) associated with the undamped wave motion of the KAW. The paper does have one major issue in that it is not made clear that what is measured is indeed just the conservative energy transfer of the undamped wave motion. (That is not to diminish the accomplishment of this important result.) In some places, it seems to imply that the energy transfer measured may be associated with a net resonant energy transfer, and that will need to be corrected.

As well, I have made a large number of comments intended at improving the paper, both clarifying arguments as well as directly providing more of the computed values (and their uncertainties) discussed in the paper.

Overall, it is an important result that demonstrates the unique capabilities of the MMS mission as well as verifies the energy transfer associated with KAWs using spacecraft measurements. Although some revisions are necessary, I foresee that this paper will be accepted upon suitable revision, and it is of sufficient impact to justify publication in Nature Communications.

=====

Major Comments:

1. Line 98: The statement

a) "... the plasma heating term, $\mathbf{J} \cdot \mathbf{E}_p$, is equal to zero throughout the entire wave."

is incorrect, and leads to some potential misinterpretation of the primary results reported in this paper. Specifically, for an Alfvén wave of amplitude B_1 in the ideal MHD theory with a wavevector $\mathbf{k} = k_z \hat{z} + k_x \hat{x}$, the current can be shown to have the form (in cgs)

$$\mathbf{j} = (k_x c B_1 / 4 \pi) \sin(k_z z + k_x x - \omega t) \hat{z} + (k_z c B_1 / 4 \pi) \sin(k_z z + k_x x - \omega t) \hat{x}$$

and the electric field

$$\mathbf{E} = - (v_A B_1 / c) \cos(k_z z + k_x x - \omega t) \hat{x}$$

Therefore, $\mathbf{j} \cdot \mathbf{E}$ is nonzero, but rather the two \hat{x} components are 90 deg out of phase. This means that there is indeed nonzero energy transfer back and forth between fields and particles associated with the undamped wave motion. This component of energy transfer is, in fact, exactly what is directly measured in this paper. Only if the current is phase shifted away from 90 deg (relative to \mathbf{E}) will the $\mathbf{j} \cdot \mathbf{E}$ be net nonzero over a wave period, corresponding to the net transfer of energy between fields and particles.

b) Line 102-104: The observations are not in contrast to the largely undamped wave dynamics of a KAW. The KAW has nonzero E_{par} , and if it is weakly damped, all components of E_{par} and E_{perp} will be approximately 90 deg phase-shifted from j_{par} and j_{perp} , respectively.

c) Line 106-107:

"the first direct measurements of local energy exchange between the particles and fields that comprise a kinetic-scale Alfvén-branch wave"

This is indeed a nice observation, but the text makes it sound like it is something other than just the conservative energy transfer back and forth between particles and fields that is associated with undamped wave motion.

d) Line 157-160: To reiterate, the observations are simply consistent with undamped KAW dynamics, and that should be made clear in the text.

2. Terminology: Propagation. There is often confusing in the literature where it is stated that a wave "propagates" in some direction, meaning that that is the direction of the wavevector. A wave propagates in the direction of its group velocity. In many fluids, indeed the wave does propagate in the same direction as the wavevector, meaning the group velocity is in the same direction as the wavevector. But in magnetized plasmas this is often not the case, and it ends up leading to significant confusion. For example, an Alfvén wave in the MHD limit ALWAYS propagates along the local mean field (in a homogeneous system), regardless of the direction of the wavevector. For the kinetic Alfvén wave that is relevant here, since $k_{\perp} \gg k_{\parallel}$, the wave propagates largely along the parallel direction, where the ratio of the perp to the parallel part of the group velocity is $v_{\perp}/v_{\parallel} = k_{\parallel}/k_{\perp} \ll 1$. I will note the lines in which the use of this confusing terminology is misleading.

a) Line 57: Here you seem to be using parallel propagating correctly in reference to the MHD Alfvén wave, but this statement is a bit confusing since it is not completely clear whether you mean the wavevector or the direction of the group velocity.

b) Line 3: Here propagation certainly refers to the direction of the wavevector, and not the direction of the group velocity (which is why this terminology is extremely confusing).

3. Line 55: OR, not "and"

4. Line 164: "accounted for the unexpected signatures of perpendicular heating" I don't think this paper supports the explanation of any unexpected signatures of perpendicular heating. Increased perpendicular temperatures (such as the trapped electrons) do not imply heating; in fact, they simply imply adiabatic motion, in which there is zero heating.

5. Line 358: High pass filter at 0.5 Hz: How much does this affect the amplitude and phase of a signal at 1 Hz? The phase relationships play an important role in computing $\mathbf{j} \cdot \mathbf{E}$, so any processing that may alter the phases (even by a little bit), could have a dramatic

effect in the results. In this methods section, this should be addressed.

=====
=====

Minor Comments:

1. Line 32: ... no NET energy is transferred between ...
2. Line 44-45: "to confirm the direct exchange of energy between a marginally stable KAW field and plasma particles for the first time"
As noted above, the energy exchange measured here is just part of the undamped wave dynamics of a KAW, although the previous lines in the abstract appear to suggest the energy transfer measured here is instead that associated with a collisionless damping mechanism, such as Landau and cyclotron resonant interactions. This should be made more clear.
3. Line 49: The reference O'Neil 1965 treats the collisionless damping of electrostatic Langmuir waves, and does not involve KAWs. Any treatment of KAWs require an electromagnetic treatment, because the KAW is electromagnetic.
4. Line 77: Is this parallel or perpendicular ion plasma beta?
Parallel is probably the better choice to quote, because it is a natural parameter for investigating kinetic temperature anisotropy instabilities.
5. Line 82: Would be nice to quote the calculated value of ρ_i (which depends on the perp temperature).
6. Line 85: Please state in words here (referring to calculation described in Methods) how plasma frame frequency is computed.
7. Line 85: Clearly you do not know $k \rho_i$ to 3 significant figures. 1.0 is probably sufficient, but better would be to quote some variation based on the method of determination, such as $k \rho_i = 1.04 \pm 0.09$
8. Line 88-91: Please quote here the wave frequency (and damping rate) determined from the linear wave dispersion relation for KAWs for comparison to the observationally (multispacecraft?) determined plasma frame frequency.
9. Line 101: O'Neil reference is not relevant here as it does not deal with KAWs.
10. Line 105: The relation is backwards, because $j_{\perp} E_{\perp} \gg k_{\parallel} E_{\parallel}$.
11. Line 120: Hollweg a better reference than O'Neil here.
12. Line 122-124: "Cyclotron-resonant interactions require left-handed or right-handed polarized fluctuations for ions and electrons, respectively, both of which were present in the observed KAW"
This statement is written in a rather awkward way, making it unclear.

13. Line 126: Only one of the signs in the cyclotron resonant condition is correct for ions (which is correct depends on the convention adopted).
14. FIG 1: You should include a panel with the total B_x , B_y , B_z values, because just showing ΔB is missing important information.
15. FIG 3 Caption: You finally state λ_{par} here. It would be nice to have k_{perp} , k_{par} and ρ_i all stated clearly together in the main text.
16. Line 363-368: You state four methods here, but give no references. At this point, it is unclear exactly what methods were used. Was the k -filtering method or wave-telescope method used?
17. Line 372-387: Lots of discussion of parallel phase speeds being close to the Alfvén speed, but no numbers are given. Please cite number for each of these, include the measurements uncertainty for each.
18. Line 396: Clearly the components of K are not known to 3 significant figures. Also, it would be really helpful to project k onto \hat{b} , giving the parallel and perp components (normalized to ρ_i).
19. Line 402: Please provide measurement uncertainties for the plasma frame ω . This is VERY IMPORTANT. Often the value involves the subtraction of two large numbers, so 0.56 Hz is very hard to interpret without some estimate of the uncertainty. Also, please quote the parallel phase speed that is referred to here.
20. Line 402: $\omega/\omega_{ci} \sim 0.6$. Is this consistent with the linear dispersion relation results for a wavevector at 100 deg? The values from both should be stated and compared.
21. Line 414: "All three values in excellent agreement" Please cite the values derived, each with measurement uncertainty. Let the reader assess the quality of the agreement.
22. Line 441: These components of k would be easier to interpret if they were projected on \hat{b} before quoting for comparison.
23. Line 460: arbitrary wavevector (not propagation direction).

Reviewer #3 (Remarks to the Author):

Referee Report on NCOMMS-16-20851-T

"Wave-particle energy exchange directly observed in a kinetic Alfvén-branch wave" by D.J. Gershman et al.

The submitted manuscript provides a description and analysis of an interval of magnetospheric

plasma by MMS. In the selected interval, the authors provide evidence for the exchange of energy between an Alfvén wave (specifically a KAW) and the plasma particles. The work is interesting and novel, and I would recommend publication once the authors address a key question with their scientific interpretation.

My central concern with the results as presented are the exclusion of the possibility of transit time (or Barnes) damping (TTD) as the mechanism which is mediating the energy exchange. The authors list both Landau and cyclotron damping as candidate mechanisms, but neglect this third option. Quoting the textbook "Waves in Plasmas" by Stix (1992, pg 274) which the authors cite in other sections of the paper:

"Transit-time magnetic damping is one of two wave-particle interactions that may be identified with the $n=0$ terms in the susceptibility tensors. The other $n=0$ effect is, of course, Landau damping. Transit time magnetic damping comes from the interaction of the equivalent magnetic moment of a charged particle $\mu = m v_{\perp}^2 / 2B_0$, with the parallel gradient of the magnetic field."

The effects of TTD have been widely considered in many sub-disciplines of plasma physics, including laboratory (Berger et al 1958) and space (Barnes 1966, Quataert 1998) environments. I mention this mechanism not as an additional item to list and dismiss, but because it may potentially explain the "unpredicted population of electrons that are trapped between successive wave peaks by the magnetic mirror force." (manuscript, line 46).

The energy exchange described in this work, with the signature of magnetic mirroring, may very well be an observation of TTD. The work presented is novel, and of interest to several scientific communities; however, the author's claim that this energy exchange is "unpredicted by current KAW theories" (line 163) should be revisited in light of this potential explanation by offering proof that TTD does not explain the observed phenomena, or by recasting the observations as in situ measurements of TTD.

I stress that the observations as presented are indeed novel, and deserving of publication once this issue has been addressed. The question of energy exchange and eventual dissipation at kinetic scales is one of central concern to a number of areas of plasma physics (as discussed in the manuscript).

Secondly, the measurement of the orientation of the wavevector is discussed thoroughly in the methods section, but there is not sufficient context to the importance and novelty of such measurements. Significant debate has persisted over the last two decades over if $k_{\perp} > k_{\parallel}$ holds in various types of plasma turbulence (Shebalin et al 1983, Goldreich & Sridhar 1995, Schekochihin et al 2009), and in situ measurements of the orientation (Sahraoui et al 2010, Narita et al 2010, Narita et al 2016 or Horbury Wicks Chen 2012 for a review) have played a part in answering this debate. The fact that the wavevector magnetic field angle is measured to be $\sim 100^{\circ}$ rather than $\sim 90^{\circ}$ is of interest, and perhaps deserves further comment. The fact that all four methods, in particular the recent technique outlined in Bellan 2016, are in good agreement with one another, and with a KAW branch mode is definitely an interesting and significant result as well.

In general, the manuscript is well written and clear. The data is accessible to the interested reader, and the analysis techniques described do not need specialized codes beyond those typically accessible to scientists in the field. Significant attention is paid to describing the environment in which the measurement is performed, which allows the reader to determine the nature of the plasma MMS observed. The main section of the manuscript is concise, with no obvious sections to shorten.

A few minor/style comments:

- Are Alfvén wave the most ubiquitous wave mode in plasma physics? (line 29) Langmuir waves could also be described in such fashion.
- The caption of Fig. 2 (B) and the figure label (ΔB) are not in agreement.

-The MMS tetrahedron formation is described as high quality (line 406), but no numerical evaluation of this quality is given. As described in Narita and Glassmeier 2009 if the shape diverges strongly from a regular geometric shape, the error in the determination of gradients could be significant.

-Several characteristic numbers are simply reported as approximate (for example, $\sim 0.5 V_A$ in line 383 or $\omega \sim 0.56$ Hz in line 402). An estimate of the error involved in these quantities would assist the reader in the interpretation of the scientific claims, especially that the observed monochromatic wave is on the Alfvén branch rather than the magnetosonic branches.

Kristopher G Klein

Reviewer #2 Major Comments:

1. Line 98: The statement

a) "... the plasma heating term, $\mathbf{J} \cdot \mathbf{E}_p$, is equal to zero throughout the entire wave." is incorrect, and leads to some potential misinterpretation of the primary results reported in this paper. Specifically, for an Alfvén wave of amplitude B_1 in the ideal MHD theory with a wavevector $\mathbf{k} = k_z \hat{z} + k_x \hat{x}$, the current can be shown to have the form (in cgs)

$$\mathbf{j} = (k_x c B_1 / 4 \pi) \sin(k_z z + k_x x - \omega t) \hat{z} + (k_z c B_1 / 4 \pi) \sin(k_z z + k_x x - \omega t) \hat{x}$$

and the electric field

$$\mathbf{E} = - (v_A B_1 / c) \cos(k_z z + k_x x - \omega t) \hat{x}.$$

Therefore, $\mathbf{j} \cdot \mathbf{E}$ is nonzero, but rather the two \hat{x} components are 90 deg out of phase. This means that there is indeed nonzero energy transfer back and forth between fields and particles associated with the undamped wave motion. This component of energy transfer is, in fact, exactly what is directly measured in this paper. Only if the current is phase shifted away from 90 deg (relative to \mathbf{E}) will the $\mathbf{j} \cdot \mathbf{E}$

\dot{E} be net nonzero over a wave period, corresponding to the net transfer of energy between fields and particles.

b) Line 102-104: The observations are not in contrast to the largely undamped wave dynamics of a KAW. The KAW has nonzero E_{par} , and if it is weakly damped, all components of E_{par} and E_{perp} will be approximately 90 deg phase-shifted from j_{par} and j_{perp} , respectively.

c) Line 106-107:

"the first direct measurements of local energy exchange between the particles and fields that comprise a kinetic-scale Alfvén-branch wave"

This is indeed a nice observation, but the text makes it sound like it is something other than just the conservative energy transfer back and forth between particles and fields that is associated with undamped wave motion.

d) Line 157-160: To reiterate, the observations are simply consistent with undamped KAW dynamics, and that should be made clear in the text.

We thank the reviewer for identifying this issue with our explanation of the $\dot{J} \cdot E_p$ signatures. We agree with the above points and have adjusted the manuscript text to explicitly state that we are measuring the conservative energy transfer between particles and fields. We discuss that for an undamped ideal Alfvén wave, out of phase transverse \mathbf{J} and E_p fluctuations are expected and that for the undamped KAW mode, out of phase parallel \mathbf{J} and E_p fluctuations should exist. We present our results as experimental confirmation of these predicted signatures. Changes were made to the abstract, the initial discussion of $\dot{J} \cdot E_p$ in L71-90 (in the revised manuscript), in the interpretation of the fluctuations presented in L227-240, and in the conclusion.

2. Terminology: Propagation. There is often confusing in the literature where it is stated that a wave "propagates" in some direction, meaning that that is the direction of the wavevector. A wave propagates in the direction of its group velocity. In many fluids, indeed the wave does propagate in the same direction as the wavevector, meaning the group velocity is in the same direction as the wavevector. But in magnetized plasmas this is often not the case, and it ends up leading to significant confusion. For example, an Alfvén wave in the MHD limit ALWAYS propagates along the local mean field (in a homogeneous system), regardless of the direction of the wavevector. For the kinetic Alfvén wave that is relevant here, since $k_{\text{perp}} \gg k_{\text{par}}$, the wave propagates largely along the parallel direction, where the ratio of the perp to the parallel part of the group velocity is $v_{\text{perp}}/v_{\text{par}} = k_{\text{par}}/k_{\text{perp}} \ll 1$. I will note the lines in which the use of this confusing terminology is misleading.

a) Line 57: Here you seem to be using parallel propagating correctly in reference to the MHD Alfvén wave, but this statement is a bit confusing since it is not completely clear whether you mean the wavevector or the direction of the group velocity.

b) Line 3: Here propagation certainly refers to the direction of the wavevector, and not the direction of the group velocity (which is why this terminology is extremely confusing).

We have adjusted the text in the revised manuscript to refer to wavevector angle instead of propagation direction to avoid this ambiguity. We limit our use of the term 'propagation' to distinguish between the parallel and anti-parallel directions in L148-L159.

3. Line 55: OR, not "and"

The text has been adjusted to incorporate this change.

4. Line 164: "accounted for the unexpected signatures of perpendicular heating" I don't think this paper supports the explanation of any unexpected signatures of perpendicular heating. Increased perpendicular temperatures (such as the trapped electrons) do not imply heating; in fact, they simply imply adiabatic motion, in which there is zero heating.

We agree and we have revised the manuscript to more explicitly state that the increased T_{\perp} does not actually imply heating but rather indicates a non-linear trapping process of electrons. This trapping manifests itself as an apparent perpendicular heating if one only looks at the perpendicular electron temperature fluctuations.

5. Line 358: High pass filter at 0.5 Hz: How much does this affect the amplitude and phase of a signal at 1 Hz? The phase relationships play an important role in computing $\mathbf{j} \cdot \mathbf{E}$, so any processing that may alter the phases (even by a little bit), could have a dramatic effect in the results. In this methods section, this should be addressed.

The filter has very little (<1%) effect on a signal at 1 Hz in both amplitude and phase. We have included an expanded description of the filter in the methods section.

Reviewer #2 Minor Comments:

1. Line 32: ... no NET energy is transferred between ...

This change has been incorporated into the manuscript.

2. Line 44-45: "to confirm the direct exchange of energy between a marginally stable KAW field and plasma particles for the first time"
As noted above, the energy exchange measured here is just part of the undamped wave dynamics of a KAW, although the previous lines in the abstract appear to suggest the energy transfer measured here is instead that associated with a collisionless damping mechanism, such as Landau and cyclotron resonant interactions. This should be made more clear.

As discussed above, we have adjusted the manuscript to indicate that we are observing conservative energy transfer processes.

3. Line 49: The reference O'Neil 1965 treats the collisionless damping of electrostatic Langmuir waves, and does not involve KAWs. Any treatment of KAWs require an electromagnetic treatment, because the KAW is electromagnetic.

This line was removed from the revised manuscript.

4. Line 77: Is this parallel or perpendicular ion plasma beta?
Parallel is probably the better choice to quote, because it is a natural parameter for investigating kinetic temperature anisotropy instabilities.

We have included both parallel and perpendicular beta in the revised manuscript.

5. Line 82: Would be nice to quote the calculated value of ρ_i (which depends on the perp temperature).

ρ_i is now included on L119 and the equation used to calculate it is in the Methods section

6. Line 85: Please state in words here (referring to calculation described in Methods) how plasma frame frequency is computed.

With the restructured manuscript format, the calculation of plasma frame frequency has been incorporated into the main text.

7. Line 85: Clearly you do not know $k\rho_i$ to 3 significant figures.

1.0 is probably sufficient, but better would be to quote some variation based on the method of determination, such as $k\rho_i = 1.04 \pm 0.09$

We have now included the value of $k\rho_i$ and its variation with different determination techniques in the 'Wave properties' section in the Main text.

8. Line 88-91: Please quote here the wave frequency (and damping rate)

determined from the linear wave dispersion relation for KAWs for comparison to the observationally (multispacecraft?) determined plasma frame frequency.

We have included the parameters of analytical solutions of modeled KAW on L209-225 and they are shown in Figure 5.

9. Line 101: ONeil reference is not relevant here as it does not deal with KAWs.

This reference has been removed from this sentence.

10. Line 105: The relation is backwards, because $j_{\perp} E_{\perp} \gg k_{\perp} E_{\parallel}$.

Thank you for the correction! This change has been incorporated into the manuscript.

11. Line 120: Hollweg a better reference than ONeil here.

This reference has been adjusted.

12. Line 122-124: "Cyclotron-resonant interactions require left-handed or right-handed polarized fluctuations for ions and electrons, respectively, both of which were present in the observed KAW" This statement is written in a rather awkward way, making it unclear.

We have reworded this sentence.

13. Line 126: Only one of the signs in the cyclotron resonant condition is correct for ions (which is correct depends on the convention adopted).

We have removed the sign ambiguity in this condition and adopted $\omega - v_{\parallel} k_{\parallel} = \omega_{ci}$.

14. FIG 1: You should include a panel with the total B_x , B_y , B_z values, because just showing ΔB is missing important information.

We include the total field magnitude and unit vector direction such that the contribution of B_x, B_y , and B_z can be determined if of interest to the reader. In addition, B_x, B_y and B_z for all spacecraft are shown in Supplementary Figure 2.

15. FIG 3 Caption: You finally state λ_{\parallel} here. It would be nice to have k_{\perp} , k_{\parallel} and ρ_i all stated clearly together in the main text.

Wave parameters $\lambda_{\parallel}/k_{\parallel}$, k_{\perp} , and ρ_i are now all included explicitly in the main text of the manuscript.

16. Line 363-368: You state four methods here, but give no references. At this point, it is unclear exactly what methods were used. Was the k-filtering method or wave-telescope method used?

We have added references in this overview paragraph as well as in the detailed description of each method. The k-filtering/wave-telescope methods were not used as part of our analysis.

(17.-22.) Line 372-387: Lots of discussion of parallel phase speeds being close to the Alfvén speed, but no numbers are given. Please cite number for each of these, include the measurements uncertainty for each.

Line 396: Clearly the components of K are not known to 3 significant figures. Also, it would be really helpful to project k onto \hat{b} , giving the parallel and perp components (normalized to ρ_i).

Line 402: Please provide measurement uncertainties for the plasma frame ω . This is VERY IMPORTANT. Often the value involves the subtraction of two large numbers, so 0.56 Hz is very hard to interpret without some estimate of the uncertainty. Also, please quote the parallel phase speed that is referred to here.

Line 402: $\omega/\omega_{ci} \sim 0.6$. Is this consistent with the linear dispersion relation results for a wavevector at 100 deg? The values from both should be stated and compared.

Line 414: "All three values in excellent agreement" Please cite the values derived, each with measurement uncertainty. Let the reader assess the quality of the agreement.

Line 441: These components of k would be easier to interpret if they were projected on \hat{b} before quoting for comparison.

We have included additional details on the uncertainties of our determination of k and subsequent wave parameters. The linear correlation between transverse velocity and magnetic fluctuations is now explicitly quantified in L148-159. The results of phase differencing each component of B are compared explicitly in L170-182, and the uncertainty of k propagated into estimates of wavevector angle and plasma frequency are now included in L200-207. We have also included a more explicit comparison of the estimated wave parameters with those of the analytical model in L209-225).

23. Line 460: arbitrary wavevector (not propagation direction).

This change has been incorporated into the manuscript.

Reviewer #3 Comments:

My central concern with the results as presented are the exclusion of the possibility of transit time (or Barnes) damping (TTD) as the mechanism which is mediating the energy exchange. The authors list both Landau and cyclotron damping as candidate mechanisms, but neglect this third option. Quoting the textbook "Waves in Plasmas" by Stix (1992, pg 274) which the authors cite in other sections of the paper:

"Transit-time magnetic damping is one of two wave-particle interactions that may be identified with the $n=0$ terms in the susceptibility tensors. The other $n=0$ effect is, of course, Landau damping. Transit time magnetic damping comes from the interaction of the equivalent magnetic moment of a charged particle $\mu = m v_{\perp}^2 / 2B_0$, with the parallel gradient of the magnetic field."

The effects of TTD have been widely considered in many sub-disciplines of plasma physics, including laboratory (Berger et al 1958) and space (Barnes 1966, Quataert 1998) environments. I mention this mechanism not as an additional item to list and dismiss, but because it may potentially explain the "unpredicted population of electrons that are trapped between successive wave peaks by the magnetic mirror force." (manuscript, line 46).

The energy exchange described in this work, with the signature of magnetic mirroring, may very well be an observation of TTD. The work presented is novel, and of interest to several scientific communities; however, the author's claim that this energy exchange is "unpredicted by current KAW theories" (line 163) should be revisited in light of this potential explanation by offering proof that TTD does not explain the observed phenomena, or by recasting the observations as in situ measurements of TTD.

We thank the reviewer for highlighting this important need for the consideration of TTD physics to our observations! Indeed the parallel gradients in magnetic field strengths lend themselves to TTD effects that could complement Landau damping effects driven the fluctuations in the parallel electric field. Because we are observing an undamped KAW, it appears that both Landau and TTD effects are weak, at least locally. This weak damping is consistent with the symmetric phase space densities around $V_{||} \sim -V_a$. It is possible, however, the trapping of electrons resulted in the non-linear saturation of TTD. We have added discussion of these effects into L284-303 as well as highlighted TTD physics in the abstract, introduction, and conclusion.

The measurement of the orientation of the wavevector is discussed thoroughly in the methods section, but there is not sufficient context to the importance and novelty of such measurements. Significant debate has persisted over the last two decades over if $k_{\perp} > k_{||}$ holds in various types of plasma turbulence (Shebalin et al 1983, Goldreich & Sridhar 1995, Schekochihin et al 2009), and in situ measurements of the orientation (Sahraoui et al 2010, Narita et al 2010, Narita et al 2016 or Horbury Wicks Chen 2012 for a review) have played a part in answering this debate. The fact that the wavevector magnetic field angle is measured to be $\sim 100^{\circ}$ rather than $\sim 90^{\circ}$ is of interest, and perhaps deserves further comment. The fact that all four methods, in particular the recent technique outlined in Bellan 2016, are in good agreement with one another, and with a KAW branch mode is definitely an interesting and significant result as well.

We have included additional discussion/references of the previous wavevector determinations in space plasmas and highlight the seemingly unusual result of wavevector angle $\sim 100^{\circ}$ in L270-283.

A few minor/style comments:

-Are Alfvén wave the most ubiquitous wave mode in plasma physics? (line 29) Langmuir waves could also be described in such fashion.

We have adjusted this sentence to read 'a ubiquitous wave mode.'

-The caption of Fig. 2 (B) and the figure label (ΔB) are not in agreement.

Thank you for the correction! We have adjusted the Figure caption accordingly.

-The MMS tetrahedron formation is described as high quality (line 406), but no numerical evaluation of this quality is given. As described in Narita and Glassmeier 2009 if the shape diverges strongly from a regular geometric shape, the error in the determination of gradients could be significant.

We have included the tetrahedron quality factor and a corresponding reference on L119.

-Several characteristic numbers are simply reported as approximate (for example, $\sim 0.5 V_A$ in line 383 or $\omega \sim 0.56$ Hz in line 402). An estimate of the error involved in these quantities would assist the reader in the interpretation of the scientific claims, especially that the observed monochromatic wave is on the Alfvén branch rather than the magnetosonic branches.

As mentioned above, we have expanded our discussion of uncertainties and included additional quantities that should assist the reader in the interpretation of our analysis.

Reviewer #2 (Remarks to the Author):

Second Review of

Wave-particle energy exchange directly observed in a kinetic Alfvén-branch wave

by Gershman et al.

The authors have improved the manuscript markedly in response to the reviewer's comments, and have adequately addressed the major concerns in my previous review.

However, in their revision of the text, they have introduced a new technically incorrect aspect to their discussion, specifically regarding their introduction of the term "irreversibility" into the text, as described below:

1) Line 80: In discussing Landau resonant interactions, they state, "These interactions, combined with an imbalance in the number of particles that are moving faster than or slower than the wave, result in irreversible plasma heating or cooling (i.e., dissipation)"

This is incorrect. The collisionless energy transfer associated with Landau resonant wave-particle interactions are indeed reversible. Boltzmann's H-Theorem tells us that an increase in entropy (which means irreversibility) can only be accomplished through collisions (see Howes et al. ApJ 651:590, 2006, Appendix B2 for a discussion of this). Landau resonant interactions are collisionless, and therefore may be reversible.

Landau damping is a transfer of energy from the fluctuating fields to individual particle motion (which typically involves microscopic kinetic energy but not bulk kinetic energy), but this transfer can go the other way in principle. In fact, the nonlinear evolution of Landau damping can lead to a quasilinear flattening, and eventual inversion, of df/dv at the resonant velocity, which then transfer energy back to the fields.

Line 270-271:

"KAWs in turbulent space plasmas are thought to account for dissipation (i.e., irreversible heating) of plasmas at kinetic scales"

Again, this is not strictly correct. KAWs can be collisionlessly damped, removing energy (reversibly) from the electromagnetic fluctuations and thereby generating fluctuations in velocity space. These velocity space fluctuations can then be smoothed out by arbitrarily weak collisions, leading to entropy increase and irreversible heating (dissipation). It may be more accurate to say that KAWs can mediate the dissipation of the turbulence through collisionless damping.

Line 282: Only in-phase fluctuations in ΔJ and ΔE_p result in such an irreversible transfer of energy from the wave-field to the plasma

particles.

Here, again, the energy transfer is reversible (no collisions again). In this case, I think you mean to say a NET transfer of energy from the wave-field to the plasma particles.

Line 351: this electron-frame electric field is conveniently used for estimates of energy dissipation, i.e., plasma heating occurs when $J \cdot (E + v \times B) > 0$.

Again, $J \cdot E$ is not dissipation in weakly collisional plasmas, although it is very widely mis-stated in the literature. The misinterpretation comes from resistive MHD, where $J \cdot E$ is dissipation (because resistivity arises microscopically from collisions).

$J \cdot E$ is energy transfer, the electromagnetic work done by the electric field on the particles, which can clearly be either positive or negative.

Summary:

With the exception of these subtle (but important) issues of irreversibility and dissipation, the manuscript is in good shape. If the editor is satisfied that the authors have addressed these issues appropriately in a second minor revision, then I will be happy to recommend publication in Nature Comm upon completion of these minor revisions (I do not need to see the manuscript again).

Reviewer #3 (Remarks to the Author):

Gershman et al 2015- Wave-particle energy exchange directly observed in a kinetic Alfvén-branch wave:

After reviewing the updated manuscript, as well as the authors' direct responses to the set of referee concerns this paper is acceptable, in my opinion, for publication in this journal. In particular, moving a significant fraction of the paper from the supplemental to the main body of text has improved the readability of the work, and made the key techniques used in this work more transparent, which will help in their eventual adoption within the community.

There are a few technical and/or terminology issues that should be considered, which I detail below, before final publication.

In several places in the text, for instance lines 79, 270, and 282-3, the authors refer to resonant interactions leading directly to irreversible heating or dissipation. Formally, the resonant interactions lead to a damping of wave energy, transferring energy from the fields to the distribution, not necessarily dissipation (heating by the production of entropy via collisions or some other irreversible process). While this may seem a pedantic distinction, for sufficiently weak collision rates, one can construct a system in which damping occurs but dissipation is arrested due to an 'anti-phase mixing' (see Schekochihin et al 2016, Journal of Plasma Physics). Thus, even for turbulent systems, evidence for damping should not necessarily be taken as evidence for (irreversible) dissipation without closer examination.

In line 217, when comparing the homogeneous, linear dispersion relation to the observed $\omega(k)$ relation, "nearby solutions" are mentioned. As the linear dispersion relation is a high-dimensionality object, dependent on not only k and ω , but a litany of other plasma parameters, the concept of "nearby" is somewhat vague. How should the reader interpret the three quoted linear solutions? Are they all on the Alfvénic dispersion surface, or do they represent other normal mode solutions (for instance, magnetosonic modes)? Would the error in the measurement of k , θ , or the other plasma parameters that are used in the linear dispersion calculation accommodate the observed value of ω ? Support for the statement "local generation of the observed KAW was not predicted by linear wave theory" should be strengthened—perhaps by plotting $\omega(k)$ observed in Fig. 5 with reasonable error bars (as I believe another referee mentioned in the previous round of comments).

When discussing transit time (or Barnes) damping, the authors point the interested reader to the textbook by Stix, which provides a clear description of the phenomena. However, it may also be useful to point to the original paper by Barnes as well that develops the damping mechanisms (Barnes 1966).

The authors claim in the discussion that the observed wave, based solely on its wavevector amplitude and angle to the mean magnetic field is "beyond the threshold of what is classified as a KAW". The author's claim may be backed up by the nonlinear features they have detailed in the previous sections, but the angle (10 deg from perpendicular) and amplitude (near $k \rho_p = 1$) are not beyond the threshold of a typical KAW (see for instance the cartoon and associated discussion in Fig. 1 of TenBarge et al 2012).

Lastly, in line 100, I believe a space is missing in "particles,and". Additionally, in lines 285 and 286, the authors have dropped the definite 'the' in front of the Landau and transit time resonance.

Daniel J. Gershman
Department of Astronomy
University of Maryland
daniel.j.gershman@nasa.gov

Re: Wave-particle energy exchange directly observed in a kinetic Alfvén-branch wave

To the editor:

We thank the reviewers for their additional comments. We have responded (red text) to specific comments and concerns (black text) below.

Thank you for the consideration of our manuscript.

Sincerely,

Daniel J. Gershman
on behalf of all co-authors

Reviewer #2

In their revision of the text, they have introduced a new technically incorrect aspect to their discussion, specifically regarding their introduction of the term "irreversibility" into the text, as described below:

1) Line 80: In discussing Landau resonant interactions, they state, "These interactions, combined with an imbalance in the number of particles that are moving faster than or slower than the wave, result in irreversible plasma heating or cooling (i.e., dissipation)"

Line 270-271:

"KAWs in turbulent space plasmas are thought to account for dissipation (i.e., irreversible heating) of plasmas at kinetic scales"

Line 282: Only in-phase fluctuations in ΔJ and ΔE_p result in such an irreversible transfer of energy from the wave-field to the plasma particles.

Line 351: this electron-frame electric field is conveniently used for estimates of energy dissipation, i.e., plasma heating occurs when $J \cdot (E + V \times B) > 0$.

We thank the reviewer for identifying this issue with terminology. The terms 'dissipation' and 'irreversible heating' have been removed from the manuscript in these locations. We emphasize only that net heating or cooling can occur at these scales.

Reviewer #3:

In several places in the text, for instance lines 79, 270, and 282-3, the authors refer to

resonant interactions leading directly to irreversible heating or dissipation. Formally, the resonant interactions lead to a damping of wave energy, transferring energy from the fields to the distribution, not necessarily dissipation (heating by the production of entropy via collisions or some other irreversible process). While this may seem a pedantic distinction, for sufficiently weak collision rates, one can construct a system in which damping occurs but dissipation is arrested due to an 'anti-phase mixing' (see Schekochihin et al 2016, Journal of Plasma Physics). Thus, even for turbulent systems, evidence for damping should not necessarily be taken as evidence for (irreversible) dissipation without closer examination.

As described above, these changes have been incorporated into the revised manuscript.

In line 217, when comparing the homogeneous, linear dispersion relation to the observed $\omega(k)$ relation, "nearby solutions" are mentioned. As the linear dispersion relation is a high-dimensionality object, dependent on not only k and ω , but a litany of other plasma parameters, the concept of "nearby" is somewhat vague. How should the reader interpret the three quoted linear solutions? Are they all on the Alfvénic dispersion surface, or do they represent other normal mode solutions (for instance, magnetosonic modes)? Would the error in the measurement of k , θ , or the other plasma parameters that are used in the linear dispersion calculation accommodate the observed value of ω ? Support for the statement "local generation of the observed KAW was not predicted by linear wave theory" should be strengthened- perhaps by plotting $\omega(k)$ observed in Fig. 5 with reasonable error bars (as I believe another referee mentioned in the previous round of comments).

We have clarified in the manuscript that the curves in Figure 5 correspond to the Alfvén-branch of the dispersion relation and that no growth was observed along the slow or fast-mode branches. The nearby solutions are all on the Alfvén-branch and are the closest solutions in $(\omega/\omega_{ci}, k\rho_i, \theta)$ -space to the measured values. We have added a shaded region in Figure 5 that corresponds to the measured parameters in $(\omega/\omega_{ci}, k\rho_i)$ -space (including uncertainty estimates) to provide this context to the reader.

When discussing transit time or Barnes) damping, the authors point the interested reader to the textbook by Stix, which provides a clear description of the phenomena. However, it may also be useful to point to the original paper by Barnes as well that develops the damping mechanisms (Barnes 1966).

Thank you for the suggestion. We have included the Barnes paper in this discussion.

The authors claim in the discussion that the observed wave, based solely on its wavevector amplitude and angle to the mean magnetic field is "beyond the threshold of what is classified as a KAW". The author's claim may be backed up by the nonlinear features they have detailed in the previous sections, but the angle (10 deg from perpendicular) and amplitude (near $k\rho_p = 1$) are not beyond the threshold of a typical KAW (see for instance the cartoon and associated discussion in Fig. 1 of TenBarge et al 2012).

We have reworded the manuscript to address this point to instead state that the observed wave mode was close to the transition point between ideal and kinetic regimes, i.e., $k_{\perp}\rho_i \approx 1$.

Lastly, in line 100, I believe a space is missing in "particles,and". Additionally, in lines 285 and 286, the authors have dropped the definite 'the' in front of the Landau and transit time resonance.

Thank you for the corrections! These changes have been incorporated into the manuscript.